# Interpretable Neural Architecture Search via Bayesian Optimisation with Weisfeiler-Lehman Kernels

**Binxin Ru**\*, **Xingchen Wan**\*, **Xiaowen Dong, Michael A. Osborne**
Machine Learning Research Group
University of Oxford, UK
`{robin, xwan, xdong, mosb}@robots.ox.ac.uk`

## Abstract

Current neural architecture search (NAS) strategies focus only on finding a single, good, architecture. They offer little insight into why a specific network is performing well, or how we should modify the architecture if we want further improvements. We propose a Bayesian optimisation (BO) approach for NAS that combines the Weisfeiler-Lehman graph kernel with a Gaussian process surrogate. Our method optimises the architecture in a highly data-efficient manner: it is capable of capturing the topological structures of the architectures and is scalable to large graphs, thus making the high-dimensional and graph-like search spaces amenable to BO. More importantly, our method affords interpretability by discovering useful network features and their corresponding impact on the network performance. Indeed, we demonstrate empirically that our surrogate model is capable of identifying useful motifs which can guide the generation of new architectures. We finally show that our method outperforms existing NAS approaches to achieve the state of the art on both closed- and open-domain search spaces.

## 1 Introduction

Neural architecture search (NAS) aims to automate the design of good neural network architectures for a given task and dataset. Although different NAS strategies have led to state-of-the-art neural architectures, outperforming human experts' design on a variety of tasks (Real et al., 2017; Zoph and Le, 2017; Cai et al., 2018; Liu et al., 2018a;b; Luo et al., 2018; Pham et al., 2018; Real et al., 2018; Zoph et al., 2018a; Xie et al., 2018), these strategies behave in a black-box fashion, which returns little design insight except for the final architecture for deployment. In this paper, we introduce the idea of *interpretable* NAS, extending the learning scope from simply the optimal architecture to interpretable features. These features can help explain the performance of networks searched and guide future architecture design. We make the first attempt at interpretable NAS by proposing a new NAS method, NAS-BOWL; our method combines a Gaussian process (GP) surrogate with the Weisfeiler-Lehman (WL) subtree graph kernel (we term this surrogate GPWL) and applies it within the Bayesian Optimisation (BO) framework to efficiently query the search space. During search, we harness the interpretable architecture features extracted by the WL kernel and learn their corresponding effects on the network performance based on the surrogate gradient information.

Besides offering a new perspective on interpratability, our method also improves over the existing BO-based NAS approaches. To accommodate the popular cell-based search spaces, which are non-continuous and graph-like (Zoph et al., 2018a; Ying et al., 2019; Dong and Yang, 2020), current approaches either rely on encoding schemes (Ying et al., 2019; White et al., 2019) or manually designed similarity metrics (Kandasamy et al., 2018), both of which are not scalable to large architectures and ignore the important topological structure of architectures. Another line of work employs graph neural networks (GNNs) to construct the BO surrogate (Ma et al., 2019; Zhang et al., 2019; Shi et al., 2019); however, the GNN design introduces additional hyperparameter tuning, and the training of the GNN also requires a large amount of architecture data, which is particularly

---

\*Equal contribution. Codes are available at `https://github.com/xingchenwan/nasbowl`

expensive to obtain in NAS. Our method, instead, uses the WL graph kernel to naturally handle the graph-like search spaces and capture the topological structure of architectures. Meanwhile, our surrogate preserves the merits of GPs in data-efficiency, uncertainty computation and automated hyperparameter treatment. In summary, our main contributions are as follows:

- We introduce a GP-based BO strategy for NAS, NAS-BOWL, which is highly query-efficient and amenable to the graph-like NAS search spaces. Our proposed surrogate model combines a GP with the WL graph kernel (GPWL) to exploit the implicit topological structure of architectures. It is scalable to large architecture cells (e.g. 32 nodes) and can achieve better prediction performance than competing methods.
- We propose the idea of *interpretable NAS* based on the graph features extracted by the WL kernel and their corresponding surrogate derivatives. We show that interpretability helps in explaining the performance of the searched neural architectures. As a singular example of concrete application, we propose a simple yet effective motif-based transfer learning baseline to warm-start search on a new image tasks.
- We demonstrate that our surrogate model achieves superior performance with much fewer observations in search spaces of different sizes, and that our strategy both achieves state-of-the-art performances on both NAS-Bench datasets and open-domain experiments while being much more efficient than comparable methods.

## 2 PRELIMINARIES

**Graph Representation of Neural Networks**  Architectures in popular NAS search spaces can be represented as an acyclic directed graph (Elsken et al., 2018; Zoph et al., 2018b; Ying et al., 2019; Dong and Yang, 2020; Xie et al., 2019), where each graph node represents an operation unit or layer (e.g. a `conv3×3-bn-relu` in Ying et al. (2019)) and each edge defines the information flow from one layer to another. With this representation, NAS can be formulated as an optimisation problem to find the directed graph and its corresponding node operations (i.e. the directed attributed graph $G$) that give the best architecture validation performance $y(G)$: $G^* = \arg\max_G y(G)$.

**Bayesian Optimisation and Gaussian Processes**  To solve the above optimisation, we adopt BO, which is a query-efficient technique for optimising a black-box, expensive-to-evaluate objective (Brochu et al., 2010). BO uses a statistical surrogate to model the objective and builds an acquisition function based on the surrogate. The next query location is recommended by optimising the acquisition function which balances the exploitation and exploration. We use a GP as the surrogate model in this work, as it can achieve competitive modelling performance with small amount of query data (Williams and Rasmussen, 2006) and give analytic predictive posterior mean $\mu(G_t|\mathcal{D}_{t-1})$ and variance $k(G_t, G'_t|\mathcal{D}_{t-1})$ on the heretofore unseen graph $G_t$ given $t-1$ observations: $\mu(G_t|\mathcal{D}_{t-1}) = \mathbf{k}(G_t, G_{1:t-1})\mathbf{K}_{1:t-1}^{-1}\mathbf{y}_{1:t-1}$ and $k(G_t, G'_t|\mathcal{D}_{t-1}) = k(G_t, G'_t) - \mathbf{k}(G_t, G_{1:t-1})\mathbf{K}_{1:t-1}^{-1}\mathbf{k}(G_{1:t-1}, G'_t)$ where $G_{1:t-1} = \{G_1, \ldots, G_{t-1}\}$ and $\mathbf{y}_{1:t-1} = [y_1, \ldots, y_{t-1}]^{\mathrm{T}}$ are the $t-1$ observed graphs and objective function values, respectively, and $\mathcal{D}_{t-1} = \{G_{1:t-1}, \mathbf{y}_{1:t-1}\}$. $[\mathbf{K}_{1:t-1}]_{i,j} = k(G_i, G_j)$ is the $(i,j)$-th element of Gram matrix induced on the $(i,j)$-th training samples by $k(\cdot, \cdot)$, the graph kernel function. We use Expected Improvement (Mockus et al., 1978) in this work though our approach is compatible with alternative choices.

**Graph Kernels**  Graph kernels are kernel functions defined over graphs to compute their level of similarity. A generic graph kernel may be represented by the function $k(\cdot, \cdot)$ over a pair of graphs $G$ and $G'$ (Kriege et al., 2020):

$$k(G, G') = \langle \phi(G), \phi(G') \rangle_{\mathcal{H}} \tag{2.1}$$

where $\phi(\cdot)$ is some feature representation of the graph extracted by the graph kernel and $\langle \cdot, \cdot \rangle_{\mathcal{H}}$ denotes inner product in the associated reproducing kernel Hilbert space (RKHS) (Nikolentzos et al., 2019; Kriege et al., 2020). For more detailed reviews on graph kernels, the readers are referred to Nikolentzos et al. (2019), Ghosh et al. (2018) and Kriege et al. (2020).

**Algorithm 1** NAS-BOWL Algorithm.

Optional steps of the exemplary use of motif-based warm starting (Sec 3.2) are marked in *gray italics*.

1: **Input:** Maximum BO iterations $T$, BO batch size $b$, acquisition function $\alpha(\cdot)$, initial observed data on the target task $\mathcal{D}_0$, Optional: past-task query data $\mathcal{D}_{\text{past}}$ and surrogate $\mathcal{S}_{\text{past}}$
2: **Output:** The best architecture $G_T^*$
3: Initialise the GPWL surrogate $\mathcal{S}$ with $\mathcal{D}_0$
4: **for** $t = 1, \ldots, T$ **do**
5:     **if** *Pruning based on the past-task motifs* **then**
6:       *Compute the motif importance scores (equation 3.4) with $\mathcal{S}_{\text{past}}/\mathcal{S}$ on $\mathcal{D}_{\text{past}}/\mathcal{D}_t$*
7:       **while** $|\mathcal{G}_t| < B$ **do**
8:         *Generate a batch of candidate architectures and reject those which contain none of the top $25\%$ good motifs (similar procedure as Fig. 2(a))*
9:       **end while**
10:     **else**
11:       Generate $B$ candidate architectures $\mathcal{G}_t$
12:     **end if**
13:     $\{G_{t,i}\}_{i=1}^b = \arg\max_{G \in \mathcal{G}_t} \alpha_t(G|\mathcal{D}_{t-1})$
14:     Evaluate their validation accuracy $\{y_{t,i}\}_{i=1}^b$
15:     $\mathcal{D}_t \leftarrow \mathcal{D}_{t-1} \cup (\{G_{t,i}\}_{i=1}^B, \{y_{t,i}\}_{i=1}^b)$
16:     Update the surrogate $\mathcal{S}$ with $\mathcal{D}_t$
17: **end for**
18: Return the best architecture seen so far $G_T^*$

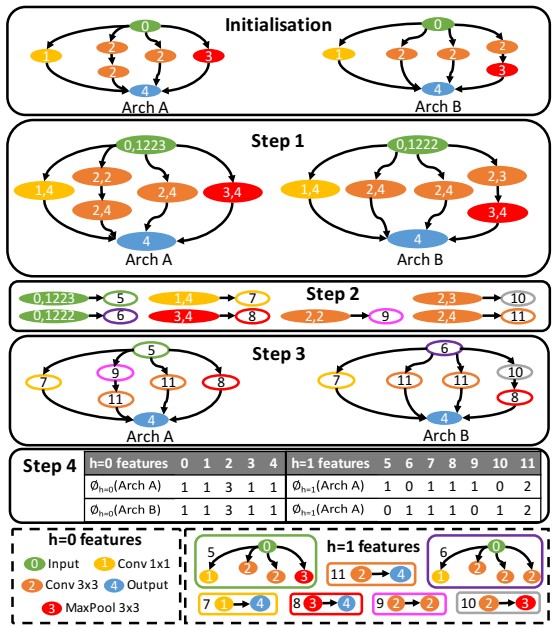

Figure 1: Illustration of one WL iteration. Given two architecture cells at initialisation, WL kernel first collects the neighbourhood labels of each node (Step 1) and compress the collected $h = 0$ labels into $h = 1$ features (Step 2). Each node is then relabelled with $h = 1$ features (Step 3) and the two graphs are compared based on the histogram on both $h = 0$ and $h = 1$ features (Step 4). This WL iteration will be repeated until $h = H$.

## 3 PROPOSED METHOD

We begin by presenting our proposed algorithm, NAS-BOWL in Algorithm 1, where there are a few key design features, namely the design of the GP surrogate suitable for architecture search (we term the surrogate GPWL) and the method to generate candidate architectures at each BO iteration. We will discuss the first one in Section 3.1. For architecture generation, we either generate the new candidates via *random sampling* the adjacency matrices, or use a *mutation algorithm* similar to those used in a number of previous works (Kandasamy et al., 2018; Ma et al., 2019; White et al., 2019; Shi et al., 2019): at each iteration, we generate the architectures by mutating a number of queried architectures that perform the best. Generating candidate architectures in this way enables us to exploit the prior information on the best architectures observed so far to explore the large search space more efficiently. We report NAS-BOWL with both strategies in our experiments. Finally, to give a demonstration of the new possibilities opened by our work, we give an exemplary practical use of intepretable motifs for transfer learning in Algorithm 1, which is elaborated in Sec 3.2.

### 3.1 SURROGATE AND GRAPH KERNEL DESIGN

To enable the GP to work effectively on the graph-like architecture search space, selecting a suitable kernel function is arguably the most important design decision. We propose to use the Weisfeiler-Lehman (WL) graph kernel (Shervashidze et al., 2011) to enable the direct definition of a GP surrogate on the graph-like search space. The WL kernel compares two directed graphs based on both local and global structures. It starts by comparing the node labels of both graphs via a base kernel $k_{\text{base}}\big(\phi_0(G), \phi_0(G')\big)$ where $\phi_0(G)$ denotes the histogram of features at level $h = 0$ (i.e. node features) in the graph, where $h$ is both the index of WL iterations and the depth of the subtree features extracted. For the WL kernel with $h > 0$, as shown in Fig. 1, it then proceeds to collect features at $h = 1$ by aggregating neighbourhood labels, and compare the two graphs with $k_{\text{base}}\big(\phi_1(G), \phi_1(G')\big)$ based on the subtree structures of depth 1 (Shervashidze et al., 2011; Höppner and Jahnke, 2020).

The procedure then repeats until the highest iteration level $h = H$ specified and the resulting WL kernel is given by:

$$k_{\mathrm{WL}}^H(G, G') = \sum_{h=0}^{H} k_{\mathrm{base}}\big(\phi_h(G), \phi_h(G')\big). \tag{3.1}$$

In the above equation, $k_{\mathrm{base}}$ is a base kernel (such as dot product) over the vector feature embedding. As $h$ increases, the WL kernel captures higher-order features which correspond to increasingly larger neighbourhoods and features at each $h$ are concatenated to form the the final feature vector ($\phi(G) = [\phi_0(G), ..., \phi_H(G)]$). The readers are referred to App. A for more detailed algorithmic descriptions of the WL kernel.

We argue that WL is desirable for three reasons. First, in contrast to many *ad hoc* approaches, WL is established with proven successes on labelled and directed graphs, by which networks are represented. Second, the WL representation of graphs is expressive, topology-preserving yet interpretable: Morris et al. (2019) show that WL is as powerful as standard GNNs in terms of discrimination power. However, GNNs requires relatively large amount of training data and thus is more data-inefficient (we validate this in Sec. 5). Also, the features extracted by GNNs are harder to interpret compared to those by WL. Note that the WL kernel by itself only measures the similarity between graphs and does not aim to select useful substructures explicitly. It is our novel deployment of the WL procedure (App. A) for the NAS application that leads to the extraction of interpretable features while comparing different architectures. We further make smart use of these network features to help explain the architecture performance in Sec 3.2. Finally, WL is efficient and scalable: denoting $\{n, m\}$ as the number of nodes and edges respectively, computing the Gram matrix on $N$ training graphs may scale $\mathcal{O}(NHm + N^2Hn)$ (Shervashidze et al., 2011). As we show in App. E.3, in typical cell-based spaces $H \leq 3$ suffices, suggesting that the kernel computation cost is likely eclipsed by the $\mathcal{O}(N^3)$ scaling of GP we incur nonetheless. This is to be contrasted to approaches such as path encoding in White et al. (2019), which scales exponentially with $n$ without truncation, and the edit distance kernel in Jin et al. (2019), whose exact solution is NP-complete (Zeng et al., 2009).

With the above-mentioned merits, the incorporation of the WL kernel permits the usage of GP-based BO on various NAS search spaces. This enables the practitioners to harness the rich literature of GP-based BO methods on hyperparameter optimisation and redeploy them on NAS problems. Most prominently, the use of GP surrogate frees us from hand-picking the WL hyperparameter $H$ as we can automatically learn the optimal values by maximising the Bayesian marginal likelihood. As we will justify in Sec. 5 and App. E.3, this process is extremely effective. This renders a further major advantage of our method as it has *no inherent hyperparameters that require manual tuning*. This reaffirms with our belief that a practical NAS method itself should require minimum tuning, as it is almost impossible to run traditional hyperparameter search given the vast resources required. Other enhancements, such as improving the expressiveness of the surrogate by combining multiple types of kernels, are briefly investigated in App. C. We find the amount of performance gain depends on the NAS search space and a WL kernel alone suffices for common cell-based spaces.

## 3.2 INTERPRETABLE NAS

The unique advantage of the WL kernel is that it extracts interpretable features, i.e. network motifs from the original graphs. This in combination with our GP surrogate enables us to predict the effect of the extracted features on the architecture performance directly by examining the derivatives of the GP predictive mean w.r.t. the features. Derivatives as tools to interpret ML models have been used previously (Engelbrecht et al., 1995; Koh and Liang, 2017; Ribeiro et al., 2016) but, given the GP, we can compute these derivatives *analytically*. Following the notations in Sec. 2, the derivative with respect to $\phi^j(G_t)$, the $j$-th element of $\phi(G_t)$ (the feature vector of a graph $G_t$) is Gaussian with an expected value:

$$\mathbb{E}_{p(y|G_t, \mathcal{D}_{t-1})}\Big[\frac{\partial y}{\partial \phi^j(G_t)}\Big] = \frac{\partial \mu}{\partial \phi^j(G_t)} = \frac{\partial \langle \phi(G_t), \Phi_{1:t-1}\rangle}{\partial \phi^j(G_t)} \mathbf{K}_{1:t-1}^{-1} \mathbf{y}_{1:t-1} \tag{3.2}$$

where $\Phi_{1:t-1} = [\phi(G_1), \ldots, \phi(G_{t-1})]^{\mathrm{T}}$ is the feature matrix stacked from the feature vectors of the previous observations. Intuitively, since each $\phi^j(G_t)$ denotes the count of a WL feature in $G_t$, its derivative naturally encodes the direction and sensitivity of the objective (in this case the predicted validation accuracy) about that feature. Computationally, since the costly term, $\mathbf{K}_{1:t-1}^{-1}\mathbf{y}_{1:t-1}$, is already computed in the posterior mean, the derivatives can be obtained at minimal additional cost.

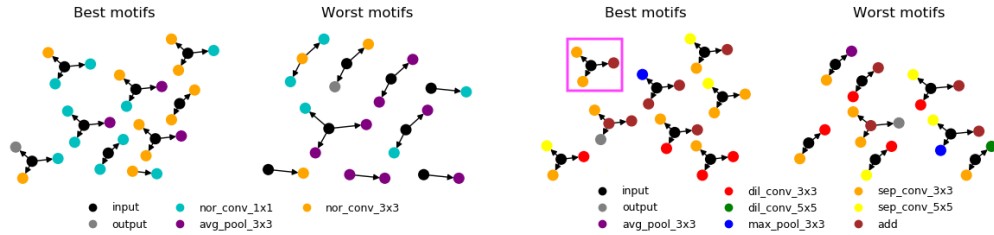

(a) Best and worst motifs identified on N201 (CIFAR-10) dataset using 300 training samples (left) and on DARTS search space after 3 GPU days of search by NAS-BOWL (right). For DARTS space, the motif boxed in pink is featured in all optimal cells found by various NAS methods in Fig 3.

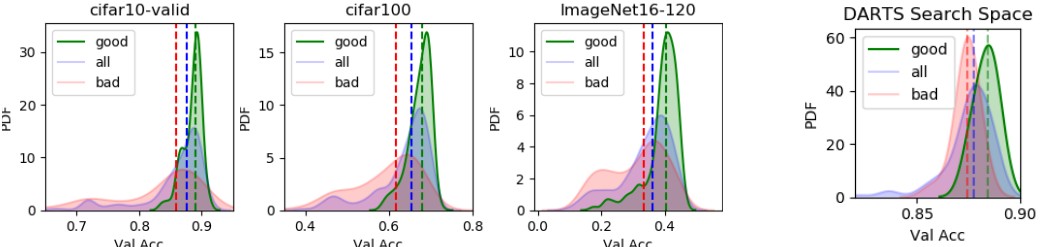

(b) Validation accuracy distributions of the validation architectures on *different tasks* of N201 (left 3) and DARTS (right). all denotes the entire validation set, while good/bad denote the distributions of the architectures with at least 1 best/worst motif, respectively; dashed lines denote the distribution medians. Note that in all cases, the good subset includes the population max and in N201, the patterns solely trained on the CIFAR-10 task also transfer well to CIFAR-100/ImageNet16.

Figure 2: Motif discovery on N201 (CIFAR-10) and DARTS spaces.

By evaluating the aforementioned derivative at some graph $G$, we obtain the *local* sensitivities of the objective function around $\phi(G)$. To achieve *global* attribution of network performance w.r.t interpretable features which we are ultimately interested in, we take inspirations from the principled averaging approach featured in many gradient-based attribution methods (Sundararajan et al., 2017; Ancona et al., 2017), by computing and integrating over the aforementioned derivatives at all training samples to obtain the *averaged gradient* (AG). AG of the $j$-th feature $\phi_j$ is given by:

$$\text{AG}(\phi^j) = \mathbb{E}_G\Big[\frac{\partial \mu}{\partial \phi^j(G)}\Big] = \int_{\phi^j(G)>0} \frac{\partial \mu}{\partial \phi^j(G)} p(\phi^j(G)) d\phi^j(G). \tag{3.3}$$

Fortunately, in WL kernel, $\phi^j(\cdot) \in \mathbb{Z}^{\geq 0} \ \forall j$ and thus $p(\phi^j(\cdot))$ is discrete, the expectation integral reduces to a weighted summation over the "prior" distribution $p(\phi^j(\cdot))\forall j$. To approximate $p(\phi^j(\cdot))$, we count the number of occurrences of each feature $\phi^j(G_n)$ in all the training graphs $\{G_1, ..., G_{t-1}\}$ where $\phi^j(\cdot)$ is present and assign weights according to its frequency of occurrence. Formally, denoting $\mathcal{G}$ as the subset of the training graphs where for each of its element $\phi^j > 0$, we have

$$\text{AG}(\phi^j) \approx \frac{\sum_{n=1}^{|\mathcal{G}|} w_n(\phi_j)\frac{\partial \mu}{\partial \phi^j(G_n)}}{\sum_{n=1}^{|\mathcal{G}|} w_n(\phi_j)} \text{ where } w_n(\phi_j) = \frac{1}{|\mathcal{G}|}\sum_{n'=1}^{|\mathcal{G}|} \delta(\phi_j(G_n), \phi_j(G_{n'})) \tag{3.4}$$

where $\delta(\cdot, \cdot)$ is the Kronecker delta function and $|\cdot|$ the cardinality of a set. Finally, we additionally incorporate the uncertainty of the derivative estimation by also normalising AG with the square root of the *empirical variance* (EV) to penalise high-variance (hence less trustworthy as a whole) gradient estimates closer to 0. EV may be straightforwardly computed:

$$\text{EV}(\phi^j) = \mathbb{V}_G\Big[\frac{\partial \mu}{\partial \phi^j(G)}\Big] = \mathbb{E}_G\Big[\big(\frac{\partial \mu}{\partial \phi^j(G)}\big)^2\Big] - \Big(\mathbb{E}_G\big[\frac{\partial \mu}{\partial \phi^j(G)}\big]\Big)^2. \tag{3.5}$$

The resultant derivatives w.r.t. interpretable features $\text{AG}(\phi^j)/\sqrt{\text{EV}(\phi^j)}$ allow us to directly identify the most influential motifs on network performance. By considering the presence or absence of such motifs, we may explain the competitiveness of an architecture or the lack of it, provided the surrogate is accurate which we show is the case in Sec. 5. More importantly, beyond passive *explaining*, we can also actively use these features as building blocks to facilitate manual *construction* of promising networks, or as priors to *prune* the massive NAS search space, which we believe would be of interest to both human designers and NAS practitioners. To validate this, we train our GPWL on architectures drawn from various search spaces, rank all the features based on their computed derivatives and show the motifs with most positive and negative derivatives (hence the most and least desirable features)[1].

---

[1]For reproduciability, we include detailed procedures and selection criteria in App. D.1.

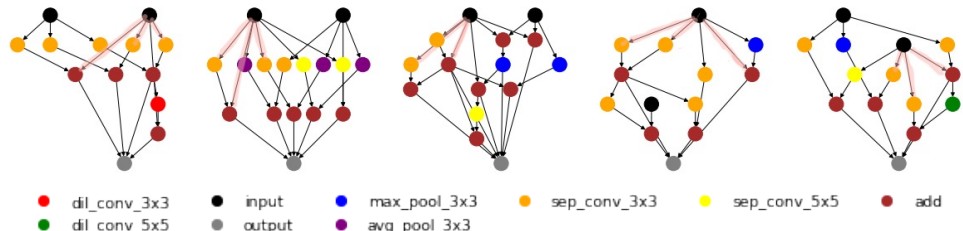

Figure 3: Best cells discovered by (left to right) DARTS, ENAS, LaNet, BOGCN and NAS-BOWL (ours) in the DARTS search space. Note the dominance of separable convolutions (especially `3x3`) in operation nodes (i.e. nodes excluding `input`, `output` and `add`) and the presence of highlighted structures encompassing the boxed motif in Fig. 2 in all cells.

We present the extracted network motifs on the CIFAR-10 task of NAS-Bench-201 (N201) (Dong and Yang, 2020) and DARTS search space in Fig. 2(a). The motifs extracted on other N201 image tasks (CIFAR-100/ImageNet16) and on NAS-Bench-101 (N101) (Ying et al., 2019) are shown in App. D.2. The results reveal some interesting insights on network performance: for example, almost every good motif in N201 contains `conv_3×3` and all-but-one good motifs in the DARTS results contain at least one *separable* `conv_3×3`. In fact, this preference over (separable) convolutions is almost universally observed in many popular NAS methods (Liu et al., 2018a; Shi et al., 2019; Wang et al., 2019; Pham et al., 2018) and ours: besides skip links, the operations in their best cells are dominated by separable convolutions (Fig. 3). Moving from node operation label to higher-order topological features, in both search spaces, our GPWL consistently finds a series of high-performing motifs that entail the parallel connection from input to multiple `conv`s often of different filter sizes – this corresponds to the *grouped convolution* unit critical to the success of, e.g. ResNeXt (Xie et al., 2017). A specific example is the boxed motif in Fig. 2(a), which combines parallel `conv`s with a skip link. This motif or other highly similar ones are consistently present in the optimal cells found by many NAS methods including ours (as shown in Fig. 3) despite the disparity in their search strategies. This suggests a correlation between these motifs and good architecture performance. Another observation is that a majority of the important motifs for both search spaces in Fig. 2(a) involve the input. From this and our previous remarks on the consensus amongst NAS methods in favouring certain operations and connections, we hypothesise that at least for a cell-based search space, the network performance might be determined more by the connections in the vicinity to the inputs (on which the optimal cells produced by different NAS methods are surprisingly consistent) than other parts of the network (on which they differ). This phenomenon is partly observed in Shu et al. (2019), where the authors found that NAS algorithms tend to favour architecture cells with most intermediate nodes having direct connection with the input nodes. The verification of this hypothesis is beyond the scope of this paper, but this shows the potential of our GPWL in discovering novel yet interpretable network features, with potential implications for both NAS and manual network design.

Going beyond the qualitative arguments above, we now quantitatively validate the informativeness of the motifs discovered. After identifying the motifs, in N201, we randomly draw another 1,000 validation architectures unseen by the surrogate. Given the motifs identified in Fig. 2(a), an architecture is labelled either "good" ($\geq 1$ good motif), "bad" ($\geq 1$ bad motif) or neither. Note if an architecture contains both good and bad motifs, it is both "good" and "bad". As demonstrated in Fig. 2(b), we indeed find that the presence of important motifs is predictive of network performance. A similar conclusion holds for DARTS space. However, due to the extreme cost in sampling the open-domain space, we make two modifications to our strategy. Firstly, the training samples are taken from a BO run, instead of randomly sampled. Secondly, we reuse the training samples for Fig. 2(b) instead of sampling and evaluating the hold-out sets. The key takeaway here is that motifs are effective in identifying promising and unpromising candidates, and thus can be used to aid NAS agents to partition the vast combinatorial search space, which is often considered a key challenge of NAS, and to focus on the most promising sub-regions. More importantly, the motifs are also *transferable*: while the patterns in Fig. 2(a) are solely trained on the CIFAR-10, they generalise well to CIFAR-100/ImageNet16 tasks – this is unsurprising, as one key motivation of cell-based search space is *exactly* to improve transferability of the learnt structure across related tasks (Zoph et al., 2018a). Given that motifs are the building blocks of the cells, we expect them to transfer well, too.

With this, we propose a simple transfer learning baseline as a singular demonstration of how motifs could be practically useful for NAS. Specifically, we can exploit the motifs identified on one task to warm-start the search on a related new task. With reference to Algorithm 1, under the transfer learning setup, we use a GPWL surrogate trained on the query data of a past related task $\mathcal{S}_{\text{past}}$ as well as the surrogate on the new target task $\mathcal{S}$ to compute the AG of motifs present in queried architectures

(equation 3.4) and identify the most positively influential motifs similar to Fig. 2(a) (**Line 3**). We then use these motifs to generate a set of candidate architectures $\mathcal{G}_t$ for optimising the acquisition function at every BO iteration on the new task; Specifically, we only accept a candidate if it contains at least one of the top $25\%$ good motifs (i.e. *pruning rule*). Finally, with more query data obtained on the target task, we will dynamically update the surrogate $\mathcal{S}$ and the motif scores to mitigate the risk of discarding motifs purely based on the past task data. Through this, we force the BO agent to select from a smaller subset of architectures deemed more promising from a previous task, thereby "warm starting" the new task. We briefly validate this proposal in the N201 experiments of Sec. 5.

## 4  RELATED WORK

In terms of NAS strategies, there have been several recent attempts in using BO (Kandasamy et al., 2018; Ying et al., 2019; Ma et al., 2019; Shi et al., 2019; White et al., 2019). To overcome the limitations of conventional BO for discrete and graph-like NAS search spaces, Kandasamy et al. (2018) use optimal transport to design a similarity measure among neural architectures while Ying et al. (2019) and White et al. (2019) suggest encoding schemes to characterise neural architectures with discrete and categorical variables. Yet, these methods are either computationally inefficient or not scalable to large architectures/cells (Shi et al., 2019; White et al., 2019). Alternatively, several works use graph neural networks (GNNs) as the surrogate model (Ma et al., 2019; Zhang et al., 2019; Shi et al., 2019) to capture the graph structure of neural networks. However, the design of the GNN introduces many additional hyperparameters to be tuned and GNN requires a relatively large number of training data to achieve decent prediction performance as shown in Sec. 5. Another related work (Ramachandram et al., 2018) apply GP-based BO with diffusion kernels to design multimodal fusion networks; however, it assigns each possible architecture as a node in an *undirected* super-graph and the need for construction of and computation on such super-graphs limits the method to relatively small search spaces. In terms of interpretability, Shu et al. (2019) study the connection pattern of network cells found by popular NAS methods and find a shared tendency for choosing wide and shallow cells which enjoy faster convergence. You et al. (2020), by representing neural networks as relational graphs, observe that the network performance depends on the clustering coefficient and average path length of its graph representation. Radosavovic et al. (2020) propose a series of manual design principles derived from extensive empirical comparison to refine a ResNet-based search space. Nevertheless, all these works do not offer a NAS strategy, and purely rely on human experts to derive insights on NAS architectures from extensive empirical studies. In contrast, our method learns the interpretable feature information without human inputs *while* searching for the optimal architecture.

## 5  EXPERIMENTS

**Surrogate Regression Performance**  We examine the regression performance of GPWL on several NAS datasets: NAS-Bench-101 (N101) on CIFAR-10 (Ying et al., 2019), and N201 on CIFAR-10, CIFAR-100 and ImageNet16. As both datasets only contain CIFAR-sized images and relatively small architecture cells[2], to further demonstrate the scalability of our proposed methods to much larger architectures, we also construct a dataset with 547 architectures sampled from the randomly wired graph generator described in Xie et al. (2019); each architecture cell has 32 operation nodes and all the architectures are trained on the Flowers102 dataset (Nilsback and Zisserman, 2008) Similar to Ying et al. (2019); Dong and Yang (2020); Shi et al. (2019), we use Spearman's rank correlation between predicted validation accuracy and the true validation accuracy as the performance metric, as what matters for comparing architectures is their relative performance ranking.

We compare the regression performance against various competitive baselines, including NASBOT (Kandasamy et al., 2018), GPs with path encodings (`PathEncode`) (White et al., 2019), GNN (Shi et al., 2019) which uses a combination of graph convolutional network and a final Bayesian linear regression layer as the surrogate, and COMBO (Oh et al., 2019)[3], which use a GP with a diffusion kernel on a graph representation of the combinatorial search spaces. We report the results in Fig. 4: our GPWL surrogate clearly outperforms all competing methods on all the NAS datasets with much

---

[2]In N101 and N201, each cell is a graph of 7 and 4 nodes, respectively.

[3]We choose COMBO as methodologically it is very close to the most related work (Ramachandram et al., 2018) whose implementation is not publicly available.

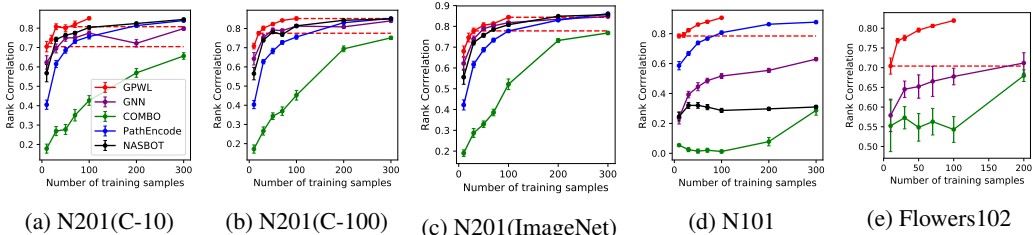

(a) N201(C-10)    (b) N201(C-100)    (c) N201(ImageNet)    (d) N101    (e) Flowers102

Figure 4: Mean Spearman correlation achieved by various surrogates across 20 trials on different datasets. Error bars denote ±1 standard error. The red dashed lines are there to help with visual comparison between the performance of GPWL and other baselines.

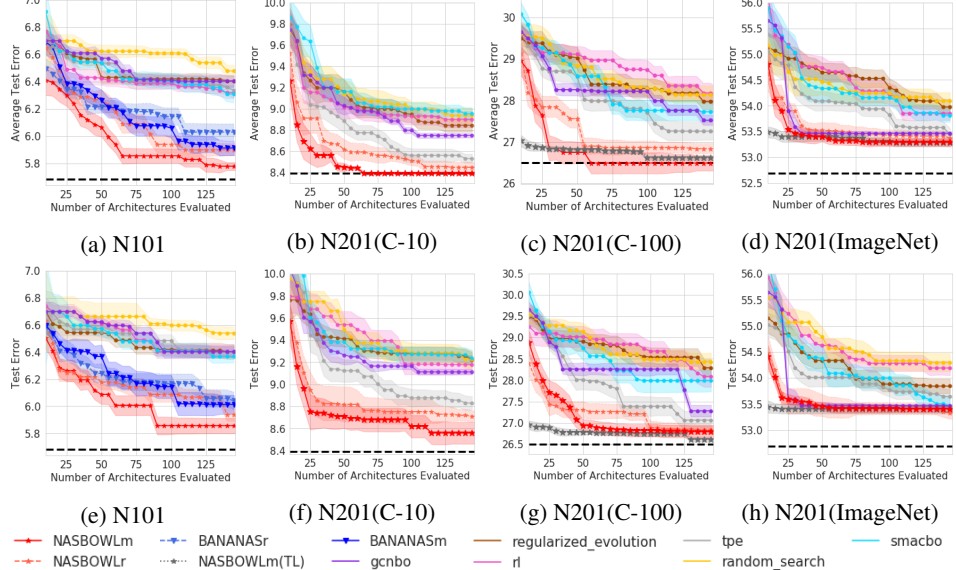

Figure 5: Median test error on NAS-Bench datasets with *deterministic* (top row) and *noisy* (bottom row) observations from 20 trials. Shades denote ±1 standard error and black dotted lines are ground-truth optima. Note the seemingly large regret in N201 (ImageNet) is due to that there are only 5 out of 15.6K architectures with test error in the interval of [52.69 (optimum), 53.25].

less training data: specifically, GPWL requires at least 3 times less data than GNN and PathEncode and 10 times less than COMBO on N201 datasets. It is also able to achieve high rank correlation on datasets with larger search spaces such as N101 and Flowers102 while requiring 20 times less data than GNN on Flowers102 and 30 times less data on N101. Moreover, in BO, uncertainty estimates are as important as the prediction accuracy; we show that GPWL produces sound uncertainty estimates in App. E.1. Finally, in addition to these surrogates previously used in NAS, we also demonstrate that our surrogate compares favourably against other popular graph kernels, as discussed in App. E.2.

**Architecture Search on NAS-Bench Datasets** We benchmark our proposed method, NAS-BOWL, against a range of existing methods, including random search, TPE (Bergstra et al., 2011), Reinforcement Learning (`rl`) (Zoph and Le, 2016), BO with SMAC (`smacbo`) (Hutter et al., 2011), regularised evolution (Real et al., 2019) and BO with GNN surrogate (`gcnbo`) (Shi et al., 2019). On N101, we also include BANANAS (White et al., 2019) which claims the state-of-the-art performance. In both NAS-Bench datasets, validation errors of different random seeds are provided, thereby creating noisy objective functions. We perform experiments using the deterministic setup described in White et al. (2019), where the validation errors over multiple seeds are averaged to eliminate stochasticity, and also report results with noisy objective functions. We show the test results in both setups in Fig. 5 and the validation results in App. F.3. In these figures, we use `NASBOWLm` and `NASBOWLr` to denote NAS-BOWL with architectures generated from mutating good observed candidates and from random sampling, respectively. Similarly, `BANANASm`/`BANANASr` represent the BANANAS with mutation/random sampling (White et al., 2019). On CIFAR-100/ImageNet tasks of N201, we also include `NASBOWLm(TL)` which is `NASBOWLm` with additional knowledge on motifs transferred from a previous run on the CIFAR-10 task of N201 to prune the candidate architectures as described in Sec. 3.2. The readers are referred to App. F.2 for detailed setups.

Table 1: Performances on CIFAR-10. GPU days do *not* include the evaluation cost of the final architecture; NAS-BOWL results from 4 random seeds on a single NVIDIA GeForce RTX 2080 Ti.

| Algorithm | Avg. Error | Best Error | #Params(M) | GPU Days |
|---|---|---|---|---|
| GP-NAS (Li et al., 2020) | - | 3.79 | 3.9 | 1 |
| DARTS(v2) (Liu et al., 2018a) | $2.76_{\pm 0.09}$ | - | 3.3 | 4 |
| ENAS[†] (Pham et al., 2018) | - | 2.89 | 4.6 | 6 |
| ASHA(Li and Talwalkar, 2019) | $3.03_{\pm 0.13}$ | 2.85 | 2.2 | 9 |
| Random-WS (Xie et al., 2019) | $2.85_{\pm 0.08}$ | 2.71 | 4.3 | 10 |
| BANANAS (White et al., 2019) | 2.64 | 2.57 | - | 12 |
| BOGCN (Shi et al., 2019) | - | 2.61 | 3.5 | 93* |
| LaNet[†] (Wang et al., 2019) | $2.53_{\pm 0.05}$ | - | 3.2 | 150 |
| **NAS-BOWL** | $\mathbf{2.61_{\pm 0.08}}$ | **2.50** | **3.7** | **3** |

†: expanded search space from DARTS. *: estimated by us. -: not reported.

It is evident that NAS-BOWL outperforms all baselines on all NAS-Bench tasks in achieving both lowest validation and test errors. The experiments with noisy observations further show that even in a more realistic setup with noisy objective function observations, NAS-BOWL still performs very well as it inherits the robustness against noise from the GP. The preliminary experiments on transfer learning also show that motifs contain extremely useful prior knowledge that may be transferred to warm-start a related task: notice that even the architectures at the very start without any search already perform well – this is particularly appealing, as in a realistic setting, searching directly on large-scale datasets like ImageNet from scratch is extremely expensive. While further experimental validation on a wider range of search spaces or tasks of varying degrees of similarity are required to fully verify the effectiveness of this particular method, we feel as an exemplary use of motifs, the promising preliminary results here already demonstrates the usefulness. Finally, we perform ablation studies in App. F.3.

**Open-domain Search**    We finally test NAS-BOWL on the open-domain search space from DARTS (Liu et al., 2018a). We allow a maximum budget of 150 queries, and we follow the DARTS setup (See App. G for details): during the search phase, instead of training the final 20-cell architectures, we train a small 8-cell architectures for 50 epochs. Whereas this results in significant computational savings, it also leads to the degraded rank correlation of performance during search and evaluation stages. This leads to a more challenging setup than most other sample-based methods which train for longer epochs and/or search on the final 20-cell architectures directly. Beyond this, we also search a single cell structure and use it for the two cell types (normal and reduction) defined in DARTS search space. We also use a default maximum number of 4 operation blocks to fit the training on a single GPU; this is contrasted to, e.g. ENAS and LaNet that allow for up to 5 and 7 blocks, respectively.

We compare NAS-BOWL with other methods in Table 1, and the best cell found by NAS-BOWL is already shown in Fig. 3 in Sec. 3.2. To ensure fairness of comparison, we only include previous methods with comparable search spaces and training techniques, and exclude methods that train much longer and/or use additional tricks (Liang et al., 2019; Cai et al., 2019). It is evident that NAS-BOWL finds very promising architectures despite operating in a more restricted setup. Consuming 3 GPU-days, NAS-BOWL is of comparable computing cost to the one-shot methods but performs on par with or better than methods that consume orders of magnitude more resources, such as LaNet which is $50\times$ more costly. Furthermore, it is worth noting that, if desired, NAS-BOWL may benefit from any higher computing budgets by relaxing the aforementioned restrictions (e.g. train longer on larger architectures during search). Finally, while we use a single GPU, NAS-BOWL can be easily deployed to run on parallel computing resources to further reduce wall-clock time.

## 6    CONCLUSION

In this paper, we propose a novel BO-based NAS strategy, NAS-BOWL, which uses a GP surrogate with the WL graph kernel. We show that our method performs competitively on both closed- and open-domain experiments with high sample efficiency. More importantly, our method represents a first step towards *interpretable NAS*, where we propose to learn interpretable network features to help explain the architectures found as well as guide the search on new tasks. The potential for further work is ample: we may extend the afforded interpretability in discovering more use-cases such as on multi-objective settings and broader search spaces. Moreover, while the current work deals primarily with *practical* NAS, we feel a thorough theoretical analysis, on e.g., convergence guarantee, would also be beneficial both for this work and the broader NAS community in general.

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

## A    ALGORITHMS

**Description of the WL kernel**    Complementary to Fig. 1 in the main tex, in this section we include a formal, algorithmic description of the WL procedure in Algorithm 2.

---

**Algorithm 2** Weisfeiler-Lehman subtree kernel computation between two graphs Shervashidze et al. (2011)

---

1: **Input:** Graphs $\{G_1, G_2\}$, Maximum WL iterations $H$
2: **Output:** The kernel function value between the graphs $k$
3: Initialise the feature vectors $\{\phi(G_1), \phi(G_2)\}$ with the respective counts of original node labels i.e. the $h = 0$ WL features. (E.g. $\phi^i(G_1)$ is the count of $i$-th node label of graph $G_1$)
4: **for** $h = 1, \ldots, H$ **do**
5:     Assign a multiset-label $M_h(v)$ to each node $v$ in $G$ consisting of the multiset $\{l_{h-1}|u \in \mathcal{N}(v)$, where $l_{h-1}(v)$ is the node label of node $v$ of the $h-1$-th WL iteration; $\mathcal{N}(v)$ are the neighbour nodes of node $v$
6:     Sort each elements in $M_h(v)$ in ascending order and concatenate them into string $s_h(v)$
7:     Add $l_{h-1}(v)$ as a prefix to $s_h(v)$.
8:     Compress each string $s_h(v)$ using hash function $f$ so that $f(s_h(v)) = f(s_h(w))$ iff $s_h(v) = s_h(w)$ for two nodes $\{v, w\}$.
9:     Set $l_h(v) := f(s_h(v)) \forall v \in G$.
10:     Concatenate the $\phi(G_1), \phi(G_2)$ with the respective counts of the *new* labels
11: **end for**
12: Compute inner product between the feature vectors in RKHS $k = \langle \phi(G_1), \phi(G_2) \rangle_{\mathcal{H}}$

---

## B    DETAILED REASONS FOR USING THE WL KERNEL

We argue that WL kernel is a desirable choice for the NAS application for the following reasons.

1. **WL kernel is able to compare labeled and directed graphs of different sizes.** As discussed in Section 2, architectures in almost all popular NAS search spaces (Ying et al., 2019; Dong and Yang, 2020; Zoph et al., 2018b; Xie et al., 2019) can be represented as directed graphs with node/edge attributes. Thus, WL kernel can be directly applied on them. On the other hand, many graph kernels either do not handle node labels (Shervashidze et al., 2009), or are incompatible with directed graphs (Kondor and Pan, 2016; de Lara and Pineau, 2018). Converting architectures into undirected graphs can result in loss of valuable information such as the direction of data flow in the architecture (we show this in Section 5).

2. **WL kernel is expressive yet highly interpretable.** WL kernel is able to capture substructures that go from local to global scale with increasing $h$ values. Such multi-scale comparison is similar to that enabled by a Multiscale Laplacian Kernel (Kondor and Pan, 2016) and is desirable for architecture comparison. This is in contrast to graph kernels such as Kashima et al. (2003); Shervashidze et al. (2009), which only focus on local substructures, or those based on graph spectra de Lara and Pineau (2018), which only look at global connectivities. Furthermore, the WL kernel is derived directly from the Weisfeiler-Lehman graph isomorphism test (Weisfeiler and Lehman, 1968), which is shown to be as powerful as a GNN in distinguishing non-isomorphic graphs (Morris et al., 2019; Xu et al., 2018). However, the higher-order graph features extracted by GNNs are hard to interpret by humans. On the other hand, the subtree features learnt by WL kernel (e.g. the $h = 0$ and $h = 1$ features in Figure **??**) are easily interpretable.

3. **WL kernel is relatively efficient and scalable.** Other expressive graph kernels are often prohibitive to compute: for example, defining $\{n, m\}$ to be the number of nodes and edges in a graph, random walk (Gärtner et al., 2003), shortest path (Borgwardt and Kriegel, 2005) and graphlet kernels (Shervashidze et al., 2009) incur a complexity of $\mathcal{O}(n^3)$, $\mathcal{O}(n^4)$ and $\mathcal{O}(n^k)$ respectively where $k$ is the maximum graphlet size. Another approach based on computing the architecture edit-distance (Jin et al., 2019) is also expensive: its exact solution is NP-complete (Zeng et al., 2009) and is provably difficult to approximate (Lin, 1994). On the other hand, the WL kernel only entails a complexity[4] of $\mathcal{O}(Hm)$ (White et al., 2019), which without truncation scales exponentially with $n$.

---

[4]Consequently, naively computing the Gram matrix consisting of pairwise kernel between all pairs in $N$ graphs is of $\mathcal{O}(N^2 Hm)$, but this can be further improved to $\mathcal{O}(NHm + N^2 Hn)$. See Morris et al. (2019).

## C    COMBINING DIFFERENT KERNELS

In general, the sum or product of valid kernels gives another valid kernel, as such, combining different kernels to yield a better-performing kernel is commonly used in GP and Multiple Kernel Learning (MKL) literature (Rasmussen, 2003; Gönen and Alpaydin, 2011). In this section, we conduct a preliminary discussion on its usefulness to GPWL. As a singular example, we consider the *additive* kernel that is a linear combination of the WL kernel and the MLP kernel:

$$k_{\mathrm{add}}(G_1, G_2) = \alpha k_{\mathrm{WL}}(G_1, G_2) + \beta k_{\mathrm{MLP}}(G_1, G_2) \text{ s.t. } \alpha + \beta = 1, \alpha, \beta \geq 0 \qquad \text{(C.1)}$$

where $\alpha, \beta$ are the kernel weights. We choose WL and MLP because we expect them to extract diverse information: whereas WL processes the graph node information directly, MLP consider the spectrum of the graph Laplacian matrix, which often reflect the global properties such as the topology and the graph connectivity. We expect the more diverse features captured by the constituent kernels will lead to a more effective additive kernel. While it is possible to determine the weights in a more principled way such as jointly optimising them in the GP log-marginal likelihood, in this example we simply set $\alpha = 0.7$ and $\beta = 0.3$. We then perform regression on NAS-Bench-101 and Flower102 datasets following the setup as in Sec. 5. We repeat each experiment 20 times and report the mean and standard deviation in Table 2, and we show the uncertainty estimate of additive kernel in Fig. 6. In both search spaces the additive kernel outperforms the constituent kernels but the gain over the WL kernel is marginal. Interestingly, while MLP performs poorly on its own, it can be seen that the complementary spectral information extracted by it can be helpful when used alongside our WL kernel. Generally, we hypothesise that as the search space increases in complexity (e.g., larger graphs, more edge connections permitted, etc), we expect that the benefits from combining different kernels to increase and we defer a more comprehensive discussion on this to a future work. As a starting point, one concrete proposal would be applying a MKL method such as ALIGNF (Cortes et al., 2012) in our context directly.

Table 2: Regression performance (i.t.o rank correlation) of additive kernels

| Kernel | N101 | Flower-102 |
|---|---|---|
| WL + MLP | **0.871**±0.02 | **0.813**±0.018 |
| WL[†] | 0.862±0.03 | 0.804±0.018 |
| MLP[†] | 0.458±0.07 | 0.492±0.12 |

†: Taken directly from Table 3.

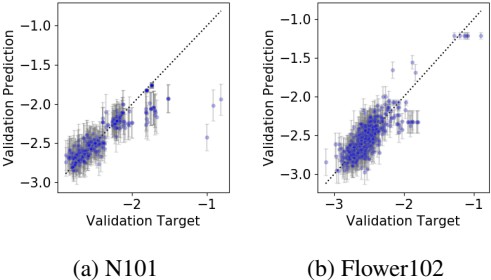

(a) N101          (b) Flower102

Figure 6: Predictive vs ground-truth validation error of GPWL with additive kernel on N101 and Flower-102 in log-log scale. Error bar denotes ±1 SD from the GP posterior predictive distribution.

## D    FURTHER DETAILS ON INTERPRETABILITY

### D.1    SELECTION PROCEDURE FOR MOTIF DISCOVERY

**NAS-Bench datasets**   In the closed-domain NAS-Bench datasets (including both NAS- bench-101 and NAS-Bench-201), we randomly sample 300 architectures from their respective search space, fit the GPWL surrogate, and compute the derivatives of all the features that appeared in the training

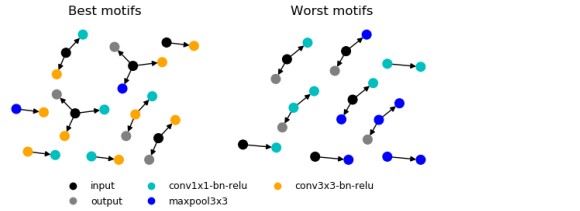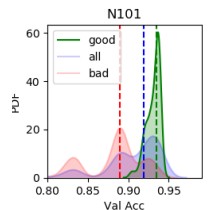

(a) Best and worst motifs identified on the N101 dataset (left), and the validation accuracy distributions in the validation architectures (right).

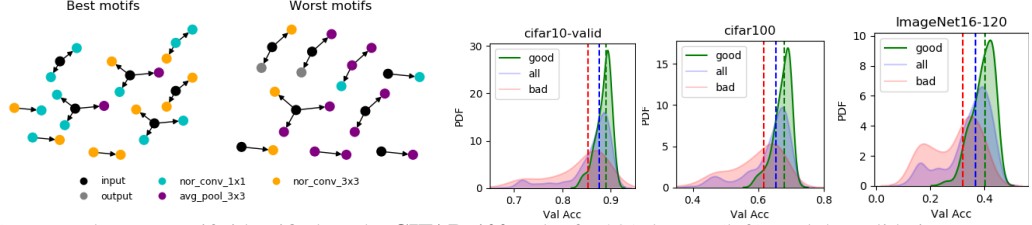

(b) Best and worst motifs identified on the **CIFAR-100** task of N201 dataset (left), and the validation accuracy distributions transferred on CIFAR-10, CIFAR-100 and ImageNet (right 3).

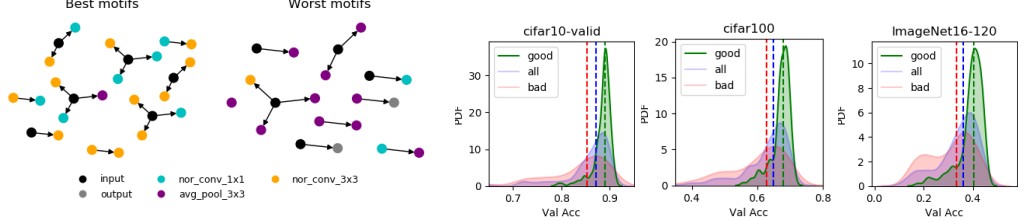

(c) Best and worst motifs identified on the **ImageNet** task of N201 dataset (left), and the validation accuracy distributions transferred on CIFAR-10, CIFAR-100 and ImageNet (right 3).

Figure 7: Motif discovery on N101 and CIFAR-100 and ImageNet tasks of N201. Note that since N101 is trained on CIFAR-10 only, it is not possible to show the results transferred on another task. All symbols and legends have the same meaning as in Fig. 2 in the main text.

set. As a regularity constraint, we then filter the motifs to only retain those that appear for more than 10 times to ensure the estimates of the derivatives by the GPWL surrogate are accurate enough and are not swayed by noises/outliers. We finally rank the features by the numerical values of the derivatives, and present the top and bottom quantiles of the features as "best motifs" and "worst motifs" respectively in Fig. 2 in the main text and Fig. 7 in Sec. D.2.

**DARTS Search Space** In the open-domain search space, it is impossible to sample efficiently since each sample drawn requires us to evaluate the architecture in full, which is computationally prohibitive. Instead, we simply reuse the GPWL surrogate trained in one run of NAS-BOWL on the open-domain experiments described in Sec. 5, which contains 120 architecture[5]-validation accuracy pair evaluated over 3 GPU days. Due to the smaller number of available samples, here we only require each feature to appear at least twice as a prerequisite, and we select the top and bottom 15% of the features to be presented in the graphs. All other treatments are identical to the descriptions above.

### D.2 MOTIF DISCOVERY ON N101 AND OTHER TASKS OF N201

Supplementary to Fig. 2 in main text, here we outline the motifs discovered by GPWL also on the N101 search space and on the other tasks (CIFAR-100, ImageNet16) of N201 in Fig. 7. We follow the identical setup as described in both the main text and Sec. D.1. In all cases, the motifs are highly effective in separating the architecture pool, and it is also noteworthy that the motifs found in the other N201 tasks are highly consistent with those shown in Fig. 2 in the main text with only minor

---

[5]The architecture here refers to the small architecture evaluated during search stage, instead of the final architecture during evaluation stage. Refer to Sec. 5 and App. G

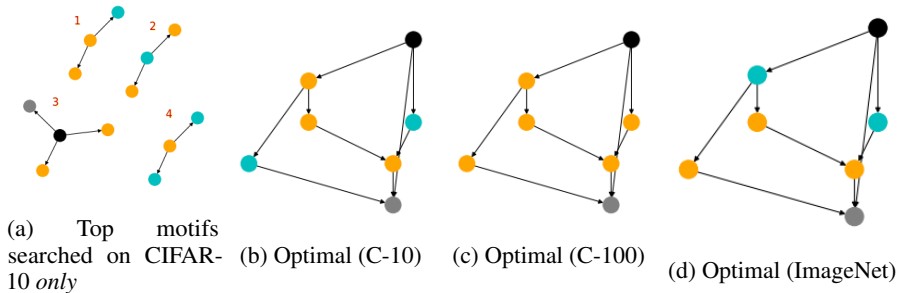

(a) Top motifs searched on CIFAR-10 *only*     (b) Optimal (C-10)     (c) Optimal (C-100)     (d) Optimal (ImageNet)

Figure 8: Computed motifs and ground-truth optimal cells for all 3 tasks of N201. Note that optimal CIFAR-10 cell contains motifs 1 and 3, optimal CIFAR-100 cell contains motif 3 and optimal ImageNet16 cell contains motif 2.

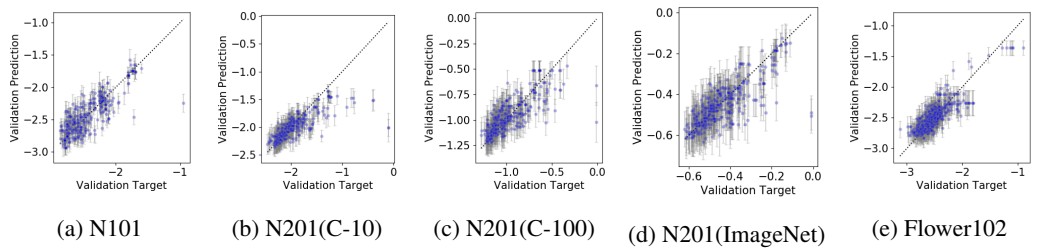

(a) N101     (b) N201(C-10)     (c) N201(C-100)     (d) N201(ImageNet)     (e) Flower102

Figure 9: Predicted vs ground-truth validation error of GPWL in various NAS-Bench tasks in log-log scale. Error bar denotes $\pm 1$ SD from the GP posterior predictive distribution.

differences, further supporting our claim that the GPWL is capable of identifying transferable features without unduly overfitting to a particular task.

To give further concrete evidence on the working and advantage of the proposed method in N201, in Fig 8 we show the top-4 motifs in terms of the derivatives computed from one experiment on CIFAR-10 *only*, according to Sec 3.2, and the ground-truth best architectures in each of the three tasks included. In this case, while the optimal cells for the different tasks are similar (but not identical) and reflective of a high-level transferability of the cells, transferring the optimal *cell* in one task directly to another will be sub-optimal. However, using our method as described in Algorithm 1 by transferring the *motifs* in Fig 8(a) on CIFAR-100 and ImageNet tasks, we reduce the search space and resultantly search time drastically (as any cell to be evaluated now needs to contain one of the motifs in Fig. 8(a)) yet we do not preemptively rule out the optimal cell (as all optimal cells contain $\geq 1$ "good" motifs). As such, our method strikes a balance between performance and efficiency.

# E  FURTHER REGRESSION RESULTS

## E.1  PREDICTIVE MEAN $\pm$ 1 STANDARD DEVIATION OF GPWL SURROGATE ON NAS DATASETS

In this section, we show the GPWL predictions on the various NAS datasets when trained with 50 samples each. It can be shown that not only a satisfactory predictive mean is produced by GPWL in terms of the rank correlation and the agreement with the ground truth, there is also sound uncertainty estimates, as we can see that in most cases the ground truths are within the error bar representing one standard deviation of the GP predictive distributions. For the training of GPWL, we always transform the validation errors (the targets of the regression) into log-scale, normalise the data and transform it back at prediction, as empirically we find this leads to better uncertainty estimates.

## E.2  COMPARISON WITH OTHER GRAPH KERNELS

We further compare the performance of WL kernel against other popular graph kernels such as (fast) Random Walk (RW) (Kashima et al., 2003; Gärtner et al., 2003), Shortest-Path (SP) (Borgwardt and Kriegel, 2005), Multiscale Laplacian (MLP) (Kondor and Pan, 2016) kernels when combined

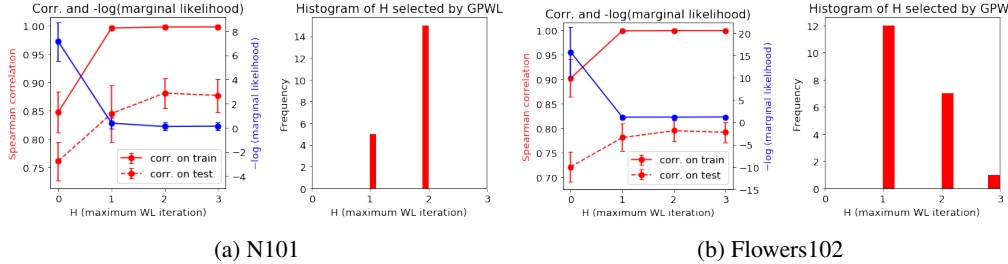

(a) N101                    (b) Flowers102

Figure 10: Spearman correlation on train/validation sets and the negative log-marginal likelihood of GP against $H$ (the maximum WL iteration) and the histograms of selected $H$ by GPWL over 20 trials on (a) N101 and (b) Flowers102.

with GPs. These competing graph kernels are chosen because they represent distinct graph kernel classes and are suitable for NAS search space with small or no modifications. In each NAS dataset, we randomly sample 50 architecture data to train the GP surrogate and use another 400 architectures as the validation set to evaluate the rank correlation between the predicted and the ground-truth validation accuracy.

We repeat each trial 20 times, and report the mean and standard error of all the kernel choices on all NAS datasets in Table 3. We also include the worst-case complexity of the kernel computation *between a pair of graphs* in the table. The results in this section justify our reasoning in App. B; combined with the interpretability benefits we discussed, WL consistently outperforms other kernels across search spaces while retaining modest computational costs. RW often comes a close competitor, but its computational complexity is worse and does not always converge. MLP, which requires us to convert directed graphs to undirected graphs, performs poorly, thereby validating that directional information is highly important.

Table 3: Regression performance (i.t.o Spearman's rank correlation) of different graph kernels.

| Kernel | Complexity | N101 | CIFAR10 | CIFAR100 | ImageNet16 | Flower-102 |
|---|---|---|---|---|---|---|
| WL | $\mathcal{O}(Hm)$ | **0.862**±0.03 | **0.812**±0.06 | **0.823**±0.03 | **0.796**±0.04 | **0.804**±0.018 |
| RW | $\mathcal{O}(n^3)$ | 0.801±0.04 | 0.809±0.04 | 0.782±0.06 | 0.795±0.03 | 0.759±0.04 |
| SP | $\mathcal{O}(n^4)$ | 0.801±0.05 | 0.792±0.06 | 0.761±0.06 | 0.762±0.08 | 0.694±0.08 |
| MLP | $\mathcal{O}(Ln^5)^{\dagger}$ | 0.458±0.07 | 0.412±0.15 | 0.519±0.14 | 0.538±0.07 | 0.492±0.12 |

†: $L$ is the number of neighbours, a hyperparameter of MLP kernel.

### E.3   VALUE OF $H$ (MAXIMUM NUMBER OF WL ITERATIONS)

As discussed, the Weisfeiler-Lehman kernel is singly parameterised by $H$, the maximum number of WL iterations. The expressive power of the kernel generally increases with $H$, as the kernel is capable of covering increasingly global features, but at the same time we might overfit into the training set, posing a classical problem of variance-bias trade-off. In this work, by combining WL with GP, we optimise $H$ against the negative log-marginal likelihood of the GP. In this section, on different data-sets we show that this approach satisfactorily balances data-fitting with model complexity.

To verify, on both N101 and Flowers102 data-sets we described, we train GPWL surrogates on 50 random training samples. On N101, we draw another 400 testing samples and on Flowers102, we use the rest of the data-set as the validation set. We use the Spearman correlation between prediction and the ground truths of the validation set as the performance metric. We summarise our result in Fig. 10: in both data-sets, we observe a large jump in performance from $H = 0$ to 1 (measured by the improvements in both validation and training Spearman correlation), and a slight dip in validation correlation from $H = 2$ to 3, suggesting an increasing amount of overfitting if we increase $H$ further. In both cases, the automatic selection described above succeeded in finding the "sweet spot" of $H = 1$ or 2, demonstrating the effectiveness of the approach.

## F   CLOSED-DOMAIN EXPERIMENTAL DETAILS

All experiments were conducted on a 36-core 2.3GHz Intel Xeon processor with 512 GB RAM.

## F.1 DATASETS

We experiment on the following datasets:

- **NAS-Bench-101** (Ying et al., 2019): The search space is an acyclic directed graph with 7 nodes and a maximum of 9 edges. Besides the `input` node and `output` node, the remaining 5 operation nodes can choose one of the three possible operations: `conv3×3-bn-relu`, `conv1×1-bn-relu` and `maxpool3×3`. The dataset contains all 423,624 unique neural architectures in the search space. Each architecture is trained for 108 epochs and evaluated on CIFAR10 image data. The evaluation is repeated over 3 random initialisation seeds. We can access the final training/validation/test accuracy, the number of parameters as well as training time of each architecture from the dataset. The dataset and its API can be downloaded from `https://github.com/google-research/nasbench/`.

- **NAS-Bench-201** (Dong and Yang, 2020): The search space is an acyclic directed graph with 4 nodes and 6 edges. Each edge corresponds to an operation selected from the set of 5 possible options: `conv1×1`, `conv3×3`, `avgpool3×3`, `skip-connect` and `zeroize`. This search space is applicable to almost all up-to-date NAS algorithms. Note although the search space of NAS-Bench-201 is more general, it's smaller than that of NAS-Bench-101. The dataset contains all 15,625 unique neural architectures in the search space. Each architecture is trained for 200 epochs and evaluated on 3 image datasets: CIFAR10, CIFAR100, ImageNet16-120. The evaluation is repeated over 3 random initialisation seeds. We can access the training accuracy/loss, validation accuracy/loss after every training epoch, the final test accuracy/loss, number of parameters as well as FLOPs from the dataset. The dataset and its API can be downloaded from `https://github.com/D-X-Y/NAS-Bench-201`.

- **Flowers102**: We generate this dataset based on the random graph generators proposed in Xie et al. (2019). The search space is an acyclic directed graph with 32 nodes and a varying number of edges. All the nodes can take one of the three possible options: `input`, `output`, `relu-conv3×3-bn`. Thus, the graph can have multiple inputs and outputs. This search space is very different from those of NAS-Bench-101 and NAS-Bench-201 and is used to test the scalability of our surrogate model for a large-scale search space (i.t.o number of numbers in the graph). The edges/wiring/connection in the graph is created by one of the three classic random graph models: Erdos-Renyi (ER), Barabasi-Albert (BA) and Watt-Strogatz (WS). Different random graph models result in graphs of different topological structures and connectivity patterns and are defined by one or two hyperparameters. We investigate a total of 69 different sets of hyperparameters: 8 values for the hyperparameter of ER model, 6 values for the hyperparameter of BA model and 55 different value combinations for the two hyperparameters of WS model. For each hyperparameter set, we generate 8 different architectures using the random graph model and train each architecture for 250 epochs before evaluating on Flowers102 dataset. The training set-ups follow Liu et al. (2019). This results in our dataset of 552 randomly wired neural architectures.

## F.2 EXPERIMENTAL SETUP

**NAS-BOWL** We use a batch size $B = 5$ (i.e., at each BO iteration, architectures yielding top 5 acquisition function values are selected to be evaluated in parallel). When mutation algorithm described in Sec. 3.2 is used, we use a pool size of $P = 200$, and half of which is generated from mutating the top-10 best performing architectures already queried and the other half is generated from random sampling to encourage more explorations in NAS-Bench-101. In NAS-Bench-201, accounting for the much smaller search space and consequently the lesser need to exploration, we simply generate all architectures from mutation. For experiments with random acquisition, we also use $P = 200$ throughout, and we also study the effect of varying $P$ later in this section. We use WL with optimal assignment (OA) (Kriege et al., 2016) for all datasets apart from NAS-Bench-201. Denoting the feature vectors of two graphs $G_1$ and $G_2$ as $\phi(G_1)$ and $\phi(G_2)$ respectively, the OA inner product in the WL case is given by the histogram intersection $\langle \phi(G_1), \phi(G_2) \rangle = \sum_j \min(\phi^j(G_1), \phi^j(G_2))$, where $\phi^j(\cdot)$ is the $j$-th element of the vector. On NAS-Bench-201 which features a much smaller search space which we find a simple dot product of the feature vectors $\phi(G_1)^T \phi(G_2)$ to perform empirically better. We always use 10 random samples to initialise NAS-BOWL.

On NAS-Bench-101 dataset, we always apply pruning (which is available in the NAS-Bench-101 API) to remove the invalid nodes and edges from the graphs. On NAS-Bench-201 dataset, since the architectures are defined over a DARTS-like, edge-labelled search space, we first convert the edge-labelled graphs to node-labelled graphs as a pre-processing step. It is worth noting that it is possible to use WL kernel defined over edge-labelled graphs directly (e.g the WL-edge kernel proposed by Shervashidze et al. (2011)), although in this paper we find the WL kernels over node-labelled graphs to perform empirically better.

On the transfer learning setup of N201, we first run a standard optimisation task on the CIFAR-10 task (we term this the *base task*) but we allow for up to an expanded budget of 250 architecture evaluations to build up the confidence of GPWL derivative estimates. We then extract the "good motifs" identified by GPWL (i.e. those features with derivatives in the top quantile). For the subsequent CIFAR-100/ImageNet16 optimisations (we term this the *transferred tasks*). On the transferred tasks, with everything else unmodified from standard runs (e.g. budget, pool size, batch size, acquisition function choice, etc), we additionally enforce the pruning rule such that only candidates in the pool with at least 1 match to the previously identified "good motifs" are allowed for evaluations and the rest are removed. The key difference is that under standard runs, the pool of size $B$ is generated *once* per BO iteration via random sampling/mutation algorithm since all candidates are accepted; here, this procedure is executed for many times as required until we have a pool of $B$ architectures where each meets the pruning criteria.

**BANANAS** We use the code made public by the authors (White et al., 2019) (`https://github.com/naszilla/bananas`), and use the default settings contained in the code with the exception of the number of architectures queried at each BO iteration (i.e. BO batch size): the default is 10, but to conform to our test settings we use 5 instead. While we do not change the default pool size of $P = 200$ at each BO iteration, instead of filling the pool entirely from mutation of the best architectures, we only mutate 100 architectures from top-10 best architectures and generate the other 100 randomly to enable a fair comparison with our method. It is worth noting that neither changes led to a significant deterioration in the performance of BANANAS: under the deterministic validation error setup, the results we report are largely consistent with the results reported in White et al. (2019); under the stochastic validation error setup, our BANANAS results actually slightly outperform results in the original paper. It is finally worth noting that the public implementation of BANANAS on NAS-Bench-201 was not released by the authors.

**GCNBO for NAS** We implemented the GNN surrogate in Sec. 5.1 by ourselves following the description in the most recent work (Shi et al., 2019), which uses a graph convolution neural network in combination with a Bayesian linear regression layer to predict architecture performance in its BO-based NAS [6]. To ensure fair comparison with our NAS-BOWL, we then define a normal Expected Improvement (EI) acquisition function based on the predictive distribution by the GNN surrogate to obtain another BO-based NAS baseline in Sec. 5.2, GCNBO. Similar to all the other baselines including our `NASBOWLr` and `BANANASr`, we use random sampling to generate candidate architectures for acquisition function optimisation. However, different from NAS-BOWL and BANANAS, GCNBO uses a batch size $B = 1$, i.e. at each BO iteration, NAS-BOWL and BANANAS select 5 new architectures to evaluate next but GCNBO select 1 new architecture to evaluate next. This setup should favour GCNBO if we measure the optimisation performance against the number of architecture evaluations which is the metric used in Figs. 4 and 5 because at each BO iteration, GCNBO selects the next architecture $G_t$ based on the most up-to-date information $\alpha_t(G|\mathcal{D}_{t-1})$ whereas NAS-BOWL and BANANAS only select one architecture $G_{t,1}$ in such fully informed way but select the other four architectures $\{G_{t,i}\}_{i=2}^5$ with outdated information. Specifically, in the sequential case ($B = 1$), $G_{t,2}$ is selected only after we have evaluated $G_{t,1}$, $G_{t,2}$ is selected by maximising $\alpha_t(G|\{\mathcal{D}_{t-1}, (G_{t,1}, y_{t,1})\})$; the same procedure applies for $G_{t,3}$, $G_{t,4}$ and $G_{t,5}$. However, in the batch case ($B = 5$) where $G_{t,i}$ for $2 \leq i \leq 5$ need to be selected before $G_{t,i-1}$ is evaluated, $\{G_{t,i}\}_{i=2}^5$ are all decided based on $\alpha_t(G|\mathcal{D}_{t-1})$ like $G_{t,1}$. For a more detailed discussion on sequential ($B = 1$) and batch ($B > 1$) BO, the readers are referred to Alvi et al. (2019).

**Other Baselines** For all the other baselines: random search (Bergstra and Bengio, 2012), TPE (Bergstra et al., 2011), Reinforcement Learning (Zoph and Le, 2016), BO with SMAC (Hutter

---

[6]Shi et al. (2019) did not publicly release their code.

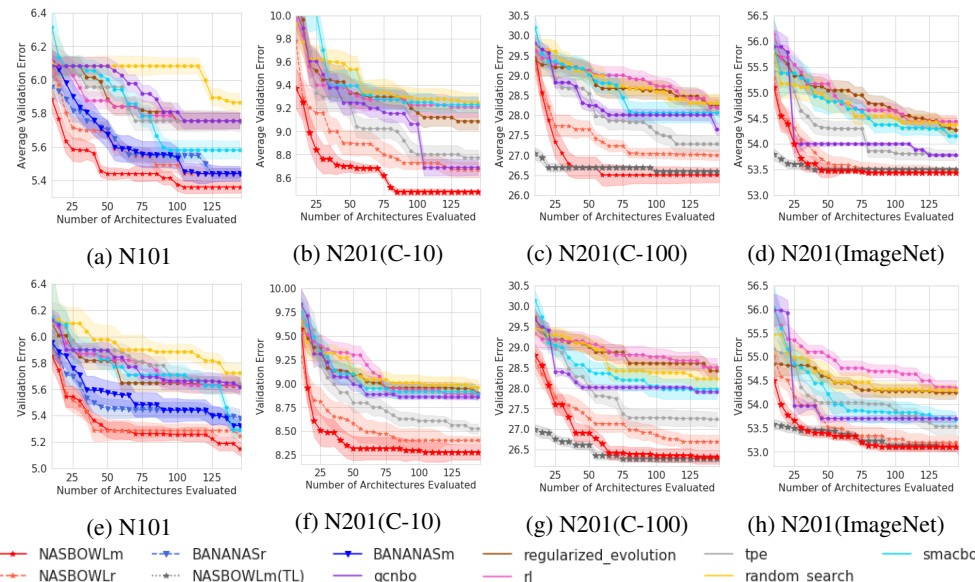

Figure 11: Median validation error on NAS-Bench datasets with *deterministic* (top row) and *noisy* (bottom row) observations from 20 trials. Shades denote $\pm 1$ standard error.

et al., 2011), regularised evolution (Real et al., 2019), we follow the implementation available at `https://github.com/automl/nas_benchmarks` for NAS-Bench-101 (Ying et al., 2019). We modify them to be applicable on NAS-Bench-201 (Dong and Yang, 2020). Note that like GCNBO, all these methods are sequential $B = 1$, and thus should enjoy the same advantage mentioned above when measured against the number of architectures evaluated.

### F.3 ADDITIONAL NAS-BENCH RESULTS

**Validation Errors Against Number of Evaluations** We show the validation errors against number of evaluations using both stochastic and deterministic validation errors of NAS-Bench datasets in Fig. 11. It is worth noting that regardless of whether the validation errors are stochastic or not, the test errors are always averaged to deterministic values for fair comparison. It is obvious that NAS-BOWL still outperforms the other methods under this metric in achieving lower test error or enjoying faster convergence, or having both under most circumstances. This corresponds well with the results on the test error in Fig. 5 and double-confirms the superior performance of our proposed NAS-BOWL in searching optimal architectures.

**Effect of Varying Pool Size** As discussed in the main text, NAS-BOWL introduces no inherent hyperparameters that require manual tuning as it relies on a non-parametric surrogate. Nonetheless, besides the surrogate, the choice on how to generate the candidate architectures requires us to specify a number of parameters such as the pool size ($P$, the number of candidate architectures to generate at each BO iteration) and batch size $B$. In our main experiments, we have set $P = 200$ and $B = 5$ throughout; in this section, we consider the effect of varying $P$ to investigate whether the performance of NAS-BOWL is sensitive to this parameter.

We keep $B = 5$ but adjust $P \in \{50, 100, 200, 400\}$, and keep all other settings to be consistent with the other experiments using the deterministic validation errors on NAS-Bench-101 (N101) (i.e. averaging the validation error seeds to remove stochasticity), and we report our results in Fig. 12 where the median result is computed from 20 experiment repeats. It can be shown that while the convergence speed varies slightly between the different $P$ choices, for all choices of $P$ apart from 50 which performs slightly worse, NAS-BOWL converges to similar validation and test errors at the end of 150 architecture evaluations – this suggests that the performance of NAS-BOWL is rather robust to the value of $P$ and that our recommendation of $P = 200$ does perform well both in terms of both the final solution returned and the convergence speed.

**Ablation Studies** In this section we perform ablation studies on the NAS-BOWL performance on both N101 and N201 (with deterministic validation errors). We repeat each experiment 20 times, and

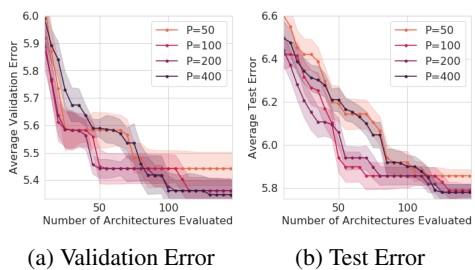

(a) Validation Error   (b) Test Error

Figure 12: Effect of varying $P$ on NAS-BOWL in N101.

we present the median and standard error in terms of both validation and test performances in Fig. 13 (N101 in (a)(b) and N201 in (c)(d)). We now explain each legend as follow:

1. `mutate`: Full NAS-BOWL with the *mutation* described in Sec. 3.2 (identical to `NASBOWLm` in Figs. 11 and 5);

2. `rand`: NAS-BOWL with *random* candidate generation. This is identical to `NASBOWLr` in Figs. 11 and 5;

3. `UCB`: NAS-BOWL with *random* candidate generation, but with the acquisition function changed from Expected Improvement (EI) to Upper Confidence Bound (UCB) Srinivas et al. (2009) $\alpha_{\mathrm{acq}} = \mu + \beta_n \sigma$, where $\mu, \sigma$ are the predictive mean and standard deviation of the GPWL surrogate, respectively and $\beta_n$ is a coefficient that changes as a function of $n$, the number of BO iterations. We select $\beta$ at initialisation ($\beta_0$) to 3, but decay it according to $\beta_n = \beta_0 \sqrt{\frac{1}{2} \log(2(n+1))}$ as suggested by Srinivas et al. (2009), where $n$ is the number of BO iterations.

4. `VH`: NAS-BOWL with *random* candidate generation, but instead of leaving the value of $h$ (number of WL iterations) to be automatically determined by the optimisation of the GP log marginal likelihood, we set $h = 0$, i.e. no WL iteration takes place and the only features we use are the counts of each type of original node operation features (e.g. `conv3×3-bn-relu`). This essentially reduces the WL kernel to a Vertex Histogram (VH) kernel.

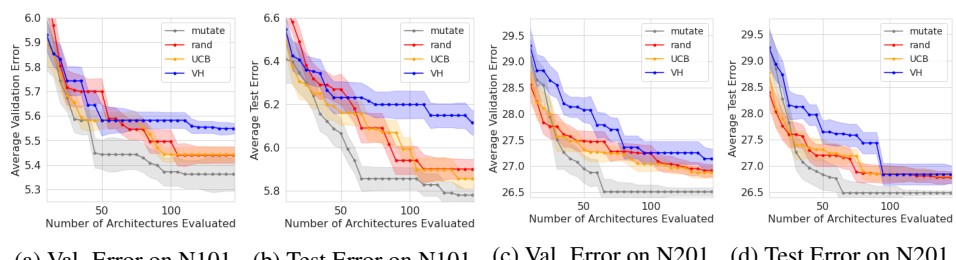

(a) Val. Error on N101   (b) Test Error on N101   (c) Val. Error on N201   (d) Test Error on N201

Figure 13: Ablation studies of NAS-BOWL

We find that topological information and using an appropriate $h$ are highly crucial: in both N101 and N201, `VH` significantly underperforms the other variants, although the extent of underperformance is smaller in N201 likely due to its smaller search space. This suggests that how the nodes are connected, which are extracted as higher-order WL features, are very important, and the multi-scale feature extraction in the WL kernel is crucial to the success of NAS-BOWL. On the other hand, the choice of the acquisition function seems not to matter as much, as there is little difference between `UCB` and `WL` runs in both N101 and N201. Finally, using mutation algorithm leads to a significant improvement in the performance of NAS-BOWL, as we have already seen in the main text.

# G   OPEN-DOMAIN EXPERIMENTAL DETAILS

All experiments were conducted on a machine with an Intel Xeon-W processor with 64 GB RAM and a single NVIDIA GeForce RTX 2080 Ti GPU with 11 GB VRAM.

## G.1 SEARCH SPACE

Our search space is identical to that of DARTS (Liu et al., 2018a): it is in the popular NAS-Net search space, and we limit the maximum number of operation nodes to be 4 (in addition to 2 input nodes and 1 input node in each cell), and the possible node operations are $3 \times 3$ and $5 \times 5$ separable convolutions (`sep-conv-3×3` and `sep-conv-5×5`), $3 \times 3$ and $5 \times 5$ dilated convolutions (`dil-conv-3×3` and `dil-conv-5×5`), $3 \times 3$ max pooling and average pooling (`max-pool-3×3` and `avg-pool-3×3`), identity skip connection (`skip-connect`) and zeroise (`none`).To enable the application of the GPWL surrogate without modification, we use the ENAS-style node-attributed DAG representation of the cells (this representation can be easily converted to the DARTS-style edge-attributed DAG without any loss of information). We show the best cell identified by NAS-BOWL in the DARTS search space in both edge- and node-attributed DAG representations in Fig, 14 as an example.

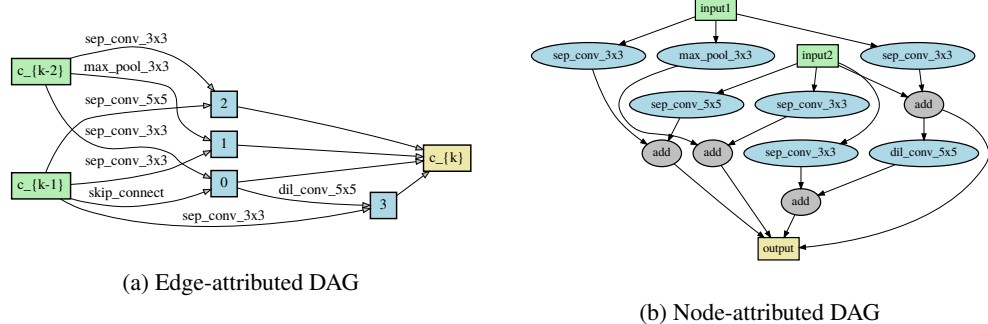

(a) Edge-attributed DAG

(b) Node-attributed DAG

Figure 14: Equivalent representations of the best cell identified by NAS-BOWL in the DARTS search space. Our method uses the node-attributed version during search, and this cell is used for both the normal and reduction cells.

## G.2 EXPERIMENTAL SETUP

We mostly follow the setup and the code base (`https://github.com/quark0/darts`) from DARTS (Liu et al., 2018a), and we detail the setup in full below:

**Architecture Search**    During the architecture search, we use half of the CIFAR-10 training data and leave the other half as the validation set. We use stack the search cell 8 times to produced a small network, and use batch size of 64 and initial number of channels of 16. As discussed, we only search one cell, and use this for both normal and reduction cells (in DARTS the two cells are searched separately). We use SGD optimiser with momentum of 0.9, weight decay of $3 \times 10^{-4}$ and an initial learning rate of 0.025 which is cosine annealed to zero over 50 epochs. As known by previous works (Liu et al., 2018a), the validation performance on CIFAR-10 is very volatile, and to ameliorate this we feed the average of the validation accuracy of the final 5 epochs as the observed accuracy to the GPWL surrogate. We use the identical setup for GPWL surrogate as the NAS-Bench experiments and we use the standard mutation algorithm described to generate the candidates every BO iteration.

**Architecture Evaluation**    After the search budget (set to 150 architectures) is exhausted, we evaluate the neural network stacked from the best architecture found, *based on the validation accuracy during the search stage*. During evaluation, we construct a larger network of 20 cells and is trained for 600 epochs with batch size 96 and initial number of channels of 36. Additional enhancements that are almost universally used in previous works, such as path dropout of probability 0.2, cutout, and auxiliary towers with weight 0.4 are also applied in this stage (these techniques are all identical to those used in Liu et al. (2018a)). Any other enhancements not used in DARTS such as mixup, AutoAugment and test-time data augmentation are not applied. The optimiser setting is identical to that during architecture search, with the exception that the cosine annealing is over the full 600 epochs instead of 50 during search. During this stage, we use the entire CIFAR-10 training set for training, and report best accuracy encountered during evaluation on the validation set in Table 1. We finally train the final architecture for 4 random repeats on CIFAR-10 dataset.

