# OpenReview forum: "Interpretable Neural Architecture Search via Bayesian Optimisation with Weisfeiler-Lehman Kernels"
_ICLR.cc/2021/Conference — ICLR 2021 Poster_

### Official Review · AnonReviewer1 · 2020-10-27
**Recommendation to strong accept**

**Rating:** 9
**Confidence:** 4

**Review:**

Summary
========
This work proposed a BO based NAS method using Weisfeiler-Lehman kernel. The idea is novel and natural considering the neural network architectures as acyclic directed graphs. I am a bit surprised to see no one tried it before in the NAS field and it is great to know that using WL kernel leads to competitive NAS performance comparing to other NAS methods and at the same time improves interpretability.

Pros
====
* The proposed idea is novel and natural, given the graph natures of network architecture.
* The notes on the interpretability is very interesting and differ the work from other methods.
* Extensive empirical studies and ablation studies.
* Extensive detail for reproducibility.
* The paper is very well written.

Minor comments
===============
I think this is a really nice work and I only have some minor comments:

* There is another line of work using BO for NAS: Ru, Binxin, Pedro Esperanca, and Fabio Carlucci. "Neural Architecture Generator Optimization." arXiv preprint arXiv:2004.01395 (2020). Would be nice to know how does the proposed method compared to it.

* The appendix C mentioned about using MKL to combine WL and MLP kernels. But in the end the author used 0.7 and 0.3 as the weights for them. I am wondering whether some simple MKL algorithm such as ALIGNF  can improve the performance here. You can find more detail in this paper: Cortes C, Mohri M, Rostamizadeh A. Algorithms for learning kernels based on centered alignment[J]. The Journal of Machine Learning Research, 2012, 13(1): 795-828.

Reason for score
==============
I liked this work a lot, it bridged NAS and BO through the usage of graph kernels (WL kernel). As a result, NAS becomes more sample efficient, which is empirically verified by extensive study in this work. The author did a very good job on the empirical evaluations, they are thorough, solid and contains many ablation studies to understand their methods.

Post rebuttal comments
=========================
I thank the authors for their responses. I encourage the authors to continue the line of work on replacing the random sampling of NAGO. Given NAS-BOWL surrogate, one can do Thompson Sampling instead of random sampling. On the MKL side, the same weights for all the kernels might be the cause of worse performance. I would also encourage the authors to verify that. Nevertheless, those are minor comments and I still think this is an important work to bridge BO and NAS. I will keep my score.

---

> ### Author Response · Authors · 2020-11-19
> **Response to Reviewer 1**
>
> Many thanks for your positive and constructive review! Please see below for our response to your comments:
>
> 1. **How does the proposed method compare to another BO-for-NAS work [1]?**
>
> We thank the reviewer for the suggestion. Neural Architecture Generator Optimization (NAGO) [1] is an interesting and relevant work but is orthogonal to our method. The key contribution of NAGO is a novel graph-generator-based **search space** which is motivated by recasting NAS as searching for an optimal architecture generator/distribution instead of an optimal architecture. On the other hand, our work proposes a new **search strategy** that can be applied to a wide variety of search spaces including that of NAGO. In fact, we’ve shown in the paper that our GPWL surrogate performs very well on the randomly wired networks [2], which inspires NAGO and has very similar architectures as those in NAGO [2]. Note that we haven’t tested our method on NAGO search space because their code is not released yet.
>
> More interestingly, we believe our method can be applied to improve NAGO: NAGO returns an optimal architecture generator but resort to random sampling to obtain the exact architecture for final use. Our NAS-BOWL can be used in place of random sampling to extract the best architecture from the good generator; One potential way is simply to use the good architecture generator learnt by NAGO to generate candidate architectures for optimising the acquisition functions in NAS-BOWL instead of using the mutation scheme. This would be an interesting future work.
>
> [1] Ru, B., Esperanca, P. and Carlucci, F., 2020. Neural Architecture Generator Optimization.Advances in Neural Information Processing Systems, 33.
>
> [2] Xie, S., Kirillov, A., Girshick, R. and He, K., 2019. Exploring randomly wired neural networks for image recognition. In Proceedings of the IEEE International Conference on Computer Vision (pp. 1284-1293).
>
> 2. **Suggestion on MKL algorithm such as ALIGNF**
>
> Many thanks for the suggestion. We have added your comment in the discussion of App. C, as a more concrete future direction. As we said, while it seems that in the present setup, using additive kernels does not improve the predictive performance by much (and MLP is quite expensive to compute), we believe leveraging on MKL literature might deliver more significant performance boosts in larger, more complicated search spaces.

---

### Official Review · AnonReviewer2 · 2020-10-27

**Rating:** 7
**Confidence:** 4

**Review:**

# Summary of the paper

The paper presents a new Bayesian optimization strategy based on Weisfeiler-Lehman Kernels for neural architecture search. The proposed method is more sample efficient than other state-of-the-art Bayesian optimization (BO) methods and allows to identify repeating motifs in well performing architectures.


Overall, I really enjoyed reading the paper and I am somewhat surprised that nobody has tried this before. The usage of the Weisfeiler-Lehman kernel is well motivated and enables users to obtain a better intuition which parts of an architecture lead to a good performance. Also, the proposed BO method outperforms other state-of-the-art BO methods across a range of competitive benchmarks. However a few points need to be clarified, and it would be great if the authors could address them in the rebuttal.


# Merits


- The paper is clearly written and the approach is well motivated

- The proposed method achieves  strong results compared to other Bayesian optimization strategies based on Gaussian process surrogate models. Also, the paper presents a thorough empirical evaluation to other state-of-the-art BO method on well established benchmarks

- As far as I know, this is the first Bayesian optimization for NAS that provides interpretable features to explain which motifs of neural networks architectures work well.



# Concerns

- Looking at Figure 12 in the appendix, it seems that the proposed method GPWL gets its main boost from the mutation strategy used  to  optimize the acquisition function. This makes me wonder whether the model is actually better than, for example BANANAS, or whether the gain in performance is mostly due to the mutation strategy? How well would BANANAS works with this mutation strategy?

- Could you also add the plots in Figure 3 with the regret on the y-axis (as in the original papers)? This would show how far away from the optimum an optimizer actually is. This is somewhat hidden with ranking plots, where an optimizer might have found an architecture with a negligible performance difference to the global optimum but has a lower rank.

- Why does start GPWL at a higher rank than the other baselines in Figure 3? And why does it stop earlier? Also the dashed red horizontal line is not explained in the caption.


# Post Rebuttal:

 I thank the authors for taking the time to address my concerns. The paper is well written and the proposed approached is promising. I therefor recommend acceptance.

---

> ### Author Response · Authors · 2020-11-19
> **Response to Reviewer 2**
>
> Many thanks for your positive and constructive review! Please see below for our response to your comments, and we hope that resolves any concerns you had:
>
> 1. **Whether the model is better than BANANAS’s… performance gain is due to the mutation strategy? Compare with BANANAS with mutation strategy...**
> We have indeed compared with BANANAS with mutation strategy (dark blue triangle, BANANASm) on NAS-Bench-101 in both Fig. 4 and 10 and show that our NAS-BOWL with mutation (NASBOWLm) outperforms that; similarly, NAS-BOWL with random strategy (NASBOWLr) also outperforms BANANAS with random (BANANASr) in the same Figs. Moreover, in our surrogate experiments in Fig, 3, we also include a baseline which combines the path-encoding scheme in BANANAS, its key contribution, with GP and shows that our GPWL surrogate achieves better predictive performance with less than ⅓ of the training data and thus is more query efficient.
> 2. **GPWL starts at a higher rank than the other baselines in Fig. 3? And why does it stop earlier? Dashed red horizontal line is not explained in the caption**.
> For all surrogate models, we start with 10 initial training data (it might be a bit difficult to discern from the figure but the lines start at 10 not 0).The high rank prediction achieved by GPWL at the start shows that GPWL can well predict the performance ranking of validation architectures after training on only 10 initial architecture data. This shows exactly the query efficiency and competitive predictive power of our GPWL which are highly valued for the real-world NAS applications. Following the practice in [1] for comparing query efficiency across different baselines, we run the other baselines for more training data than our GPWL in order to see whether they can catch up the GPWL’s performance with more training data. E.g. on NAS-Bench-201(CIFAR100) task, Path-encoding and GNN manage to achieve the similar performance (GPWL at 100 training data) with 300 training data. This again demonstrates the query efficiency of our surrogate model and it’s evident from Fig. 3 that such gain in query efficiency is larger for larger search space (NAS-Bench-101 and Flowers102). The dashed red horizontal lines are there to help comparison between the rank correlation performance of the GPWL with other baselines .
>
> [1] Ru, B., Cobb, A., Blaas, A. and Gal, Y., 2020. Bayesopt adversarial attack. In International Conference on Learning Representations.
>
> 3. **Add the regret on the y-axis in Fig. 3 (rank correlation plots) to show how far away from the optimum**
> To clarify, Fig. 3 shows the **prediction/regression** not **optimisation** performance of different surrogate models over increasing number of randomly sampled training points. We assess the predictive performance using Spearman’s rank correlation because relative performance ranking is important for comparing architectures in NAS. Fig. 8 in the App. also shows that our GPWL surrogate can predict the exact validation error very well, especially in the low error regime in which we are more interested.
> To better assess the regret, we have added in a dashed line indicating the global optimum for different NAS-Bench datasets (close-domain search spaces) for Fig. 4. If needed, we can change the y-axis from error to regret to better illustrate that.

---

### Official Review · AnonReviewer3 · 2020-10-28
**Innovative usage of WL Graph Kernel**

**Rating:** 7
**Confidence:** 3

**Review:**

The authors propose a new neural architecture search algorithm combining Bayesian optimization with the expressive and popular Weisfeiler-Lehman (WL) Graph Kernel. One advantage of using WL is the interpretable results that stem from the nature of how the kernel is computed, namely a propagation scheme through the graph. Combined the derivative of Eq. 3.2, one can extract subgraphs that are directly responsible for increased performance. In a variety of experiments, the authors show not only increased performance of detected architectures but also find subgraphs that are found by other algorithms as well.

Even though my expertise does not lie in the field of NAS, I find this work quite appealing. It is an innovative application for graph kernels, which suffer from scalability which in this setting is less of a problem. I find the aspects of novelty, interpretability, and quantitative results convincing enough to recommend acceptance. Furthermore, the work is largely well structured and written, and the figures are legible and relevant. W.r.t whether the comparison to other SOTA NAS algorithms is of good quality and fair, I think the input from reviewers with a NAS background would be valuable.

Minor comments:
•	By itself a graph kernel is a similarity measure and does not perform any subgraph selection. It happens that due to the WL propagation scheme, the WL graph kernel consists of interpretable features while computing the similarity. I would clarify this a little more in 3.1. Since this is, in my opinion, the most innovative part of the manuscript, I would even consider bringing Figure 5 from the appendix into the main paper (maybe in a condensed form). To get some space for this you could shorten some descriptions of experiment parameters and comparison methods to the appendix. Also, I think Figure 5 is not 100% complete. To make it even easier to parse, I would put boxes around the subgraphs in the $h=1 features$ box and annotate them with their respective index. E.g. the upper graph with 4 outgoing edges should be annotated with 5 and so on. This also makes it a little clearer how WL leads to larger networks in each round of propagation.

•	I would be interested in some statistics about the chosen h parameter (do you mostly find small subnetworks to lead to high performance?) and how it is being optimized (due to its discrete nature).

•	Structure: I would move 3.2 into a subsection under the experiments. In general, the manuscript reads a little squished as you have a lot of references to the appendix. It is not easy to remedy this as you don’t want the work you did go unnoticed but maybe you can leave some references out and submit a longer version of this work to a journal where you don’t suffer the space constraints of a conference paper.

•	From Table 1 it seems like the Avg. error gains are not significant as they overlap (in terms of standard deviations) for example with the DARTS results which is not bad as you still save one day of training.

---

> ### Author Response · Authors · 2020-11-19
> **Response to Reviewer 3**
>
> Many thanks for your constructive and positive feedback! Please see below for our detailed response to your comments:
>
> 1. **Some statistics about the chosen hyperparameter H and how it is being optimized**
> We optimise the hyperparameter H via Bayesian model selection; given a uniform prior over different discrete values of H, we choose the H value that leads to the maximum marginal likelihood. This approach encodes a natural Occam’s razor factor and thus automatically balances data-fitting and model complexity. And given our GP surrogate, the marginal likelihood can be computed analytically. A detailed description of the H hyperparameter statistics is shown in Fig 9 in App. and discussed in App. E.3.
>
> 2. **From Table 1 it seems the avg. error gains are not significant as they overlap with the DARTS results which is not bad as you still save one day of training**
> The performance of different search strategies on DARTS search space on CIFAR-10 tend to be very close. As shown in [1], the std of test error among 214 fully trained random architectures from DARTS search space is around 0.23%. Thus, the 0.15% drop in average test error obtained by our NAS-BOWL over DARTS is quite significant. Thanks for your appreciation on the cost-saving obtained by our method.
>
> [1] Yang, A., Esperança, P.M. and Carlucci, F.M., 2019, September. NAS evaluation is frustratingly hard. In International Conference on Learning Representations.
>
> 3. **Advice for making our novel use of WL more explicit and modifying Fig. 5**
> Thank you for the suggestions. We’ve added the point on our novel use of WL kernel in Sec. 3.1 and also amended Fig. 5. We’ll keep Fig. 5 in the App. for now to avoid the confusion over figure numbers, but we will bring it to the main text after rebuttal.
>
> 4. **Structures of the paper.** Many thanks for the suggestions and sympathizing with us on the space constraints. The position of Sec. 3.2 is indeed a bit awkward as it has a mix of results and methods (probably more evident in the revised paper now); we decide to keep it under Sec 3, as we feel interpretability is a key part of our methodology. We do share your concern and will further optimise the paper structure for a possible future journal submission.

---

### Official Review · AnonReviewer4 · 2020-10-29
**Reasonable approach, but a bit weak technical impact**

**Rating:** 5
**Confidence:** 2

**Review:**

---------- After feedback ----------

First of all, I greatly appreciate the authors patient response to me during the feedback period. The discussion was really fruitful. Unfortunately, I still have a concern about interpretability of the proposed method, which is a central topic in the paper.

> First, we find it a bit strange when the reviewer says “it is difficult to find importance/meaning of comparing motifs is unclear”, we clearly show our method does find importance in NAS-Bench-101, all 3 tasks of NAS-Bench-201 and DARTS search space (Fig 1 and 7) -- if we can't find importance/distinguish different motifs, none of the results we've shown would've been possible.

Even when a method works empirically, if a rationale behind the procedure is not clarified, a paper would not be scientifically convincing. Thus, I still do not think my claim is strange.

> Second, the example the reviewer gives is not a case when averaged gradient fails. On the contrary, it is exactly an example of when averaged gradient works. A motif with high and diverse local gradient magnitudes but average to near-0 is not important for the purpose of interpretability, as it doesn’t consistently explain the network performance by itself (just based on such motifs, one cannot conclusively deduce the impact on performance of an arbitrary, unseen architecture in general) ...

In the last response, the authors explained the interpretability issue through combination of motifs, but it did not resolve my concern. To simplify the discussion, consider a bit extreme case in which only one motif is employed in a network simultaneously, and assume WL parameter h = 0. Let g(c) = d \mu / d \phi^j |_\phi^j=c. Then, consider a hypothetical case as follows:

motif a) g(1) = 10, g(2) = 10 ... g(10) = 10, g(11) = -10, .... g(20) = -10 : AG = 0

motif b) g(1) = 1, g(2) = 1 ... g(10) = 1, g(11) = 1, .... g(20) = 1 : AG = 20

In this example, b) has a larger AG, but a) can have larger importance in practice, and now, since only one kind of motif is employed simultaneously, the explanation of the authors cannot be applied. For the exploration purpose, I do not find any rationale to consider that b) is more important than a). I know that these are extreme examples and may depend on an application scenario, but my point is that these examples reveal difficulty of interpretation of AG. The authors introduce AG at Section3.2 as an importance measure without carefully discussing how it can be interpreted in the context of the WL based exploration (just referring other papers without discussing details in a sense of the above averaging). The explanation through marginalization also does not get rid of this question. Since the interpretability is a main theme of the paper, providing a better interpretability of AG would be desired.

---------- Before feedback ----------

The paper proposes to use Weisfeiler-Lehman (WL) kernels for neural architecture search. WL kernel can incorporate the topological structure of the network, and the authors combines WL kernels with Bayesian optimization (BO) to optimize the validation performance of the network. Further, the authors also claim that WL kernels provide a useful interpretation about good / bad network structures by using the derivative of Gaussian process (GP), which can also be used for 'pruning' architectures. The performance is shown for several benchmark datasets.

Overall, the idea would be reasonable, and the approach would be useful. However, I'd have to say that the technical novelty and depth would be somewhat weak because the standard WL kernel is directly used without any significant modification, and a gradient-based importance evaluation is also a known technique (and its interpretation in this context is a bit difficult). Further, in my understanding, the paper should have provided more general discussions, not only to show data-specific observations. Detailed comments are as follows.

The proposal of the paper is not fully clear for me because the strategies are described for each one of datasets, separately. I couldn't find general procedures for the architecture search, from the main text of the paper. In practice, of-course, tuning on each dataset would be required, but showing specific tunings for well-known benchmark datasets is not attractive. A strategy applicable to wide range of tasks would be required for a methodology paper.

Interpretation of the gradient-based motif identification is difficult for me. Even when a motif has a large positive or negative gradient value, it only implies 'local' importance around the given architecture. To derive general insight, more careful treatment would be required. The authors provide the motif discovery procedure in D.1, but it should be shown in main text because interpretability is one of main theme of this paper. In Section D.1, the authors described an approach taking average of all possible values, but the average is also difficult to interpret importance because it compresses the entire space, and as the author admitted, the computation would be often intractable in practical settings.

Although the authors claim that good/bad motif identification is useful for 'pruning', no detailed general pruning procedure is shown in the main text. Providing a general algorithm would be required. What does 'prune' mean in this context? If a motif is regarded as 'bad' once, it is discarded forever, or can revive somewhere? As I mentioned above, gradient information is only local information. Even when a motif is 'bad', simply discarding it completely would be risky. Even when the 'average' gradient is used, the problem would not be mitigated, because even if the average gradient suggests a motif is useless, it may help to improve accuracy locally. For me, the rationale behind the exploration strategy with the pruning in the paper is quite unclear.

The authors claim that the identified motif is trasferable. However, evidence of this claim is not fully clear. It seems empirical suggestions only from a few (similar) datasets. When is transferring effective, how do you know it holds when, and can it harm in some case? I think that a general discussion is missing in the paper.

Another difficulty of the gradient-based importance evaluation is that the lack of uncertainty evaluation. The gradient (3.2) is the expected value of the predictive distribution of GP. Therefore, the variance is not considered. For example, if GP does not have any observations, the expected gradient would be 0 (when prior is f(x) = 0 with the unit variance for any x, which is a standard setting), but variance of gradient would be large, meaning that a motif is still has a potential to become important. Again, discarding a motif by the expected gradient without considering uncertainty is seemingly risky, though the paper lacks this kind of discussion on uncertainty, though the uncertainty evaluation is a central issue on in the context of BO.

---

> ### Author Response · Authors · 2020-11-19
> **Response to Reviewer 4 (1/3)**
>
> Many thanks for your constructive and insightful feedback! We have incorporated many of your suggestions in the updated paper, and please see below for our detailed response and we hope that would resolve any concerns you had:
>
> 1. **the gradient of a motif only implies 'local' importance around a given architecture... average is also difficult to interpret  because it compresses the entire space and the computation would be often intractable...**
>
> We agree that the procedure in App. D.1 is an important part of the paper, and have moved it to Sec. 3.2 now.
>
> [Local vs Averaged Gradients] The local gradient describes the local effect of a motif and is specific to a particular architecture. By taking the average of local gradients of a motif, we marginalise over all architectures containing the motif and obtain a global measure of importance regarding the motif feature. In fact, the integration or averaging of some sort over local gradients, as we did, is also an approach taken in many principled gradient-based global attribution methods, such as Occlusion-1 [1] and Integrated Gradient [2], to analyse the contribution of input features to the model prediction. To the best of our knowledge, we are the first in introducing this technique in combination with interpretable features to explain the NAS performance.
>
> [Tractability of Averaging] Computing motifs is actually tractable in a practical setting – as this only involves averaging over all training/queried data, and the number of queries tend to be small under the BO, and especially the NAS setting. If the reviewer is referring to our comment in App. D.1 on intractability of sampling in DARTS search space, we would like to clarify that this intractability is about **assessing** the usefulness of the motifs on a held-out validation set of 1,000 architectures sampled, which would require us to fully train and evaluate all these 1,000 architectures. NAS-Bench datasets already did this expensive evaluation for us so we can do such assessment instantly. Note the gradient computation and motifs identification only depend on the surrogate model and don’t **need** additional expensive and extensive architecture evaluations mentioned above.
>
> Furthermore, on an empirical note, we have shown quantitatively that we can obtain valuable information on the network performance by simply analysing the motifs contained without the need to actually evaluating the networks (Fig 1), and qualitatively our approach also sheds lights into the existence of sub-structures which correlate with good network performance and are implicitly picked up by various search strategies in previous works (Fig. 2). We hope these empirical evidence also supports the effectiveness of our proposed motif assessment criterion.
>
> [1] Ancona, M., Ceolini, E., Öztireli, C. and Gross, M., 2018. Towards better understanding of gradient-based attribution methods for deep neural networks. In International Conference on Learning Representations.
>
> [2] Sundararajan, M., Taly, A. and Yan, Q., 2017. Axiomatic attribution for deep networks. International Conference on Machine Learning, PMLR 70.
>
> 2. **the general pruning procedure is missing in the main text. What does 'prune' mean in this context? If a motif is regarded as 'bad' once, it is discarded forever, or can revive somewhere?**
>
> We have added the algorithm for pruning in Algorithm 1.
>
> The number of possible architectures contained in the search space is often huge. Pruning here refers to defining and applying the constraints on such a large search space to obtain a much smaller, yet promising, pool of candidate architectures, thus speeding up the search. In our case, we transfer motifs learnt from a previous, related task as well as the new target task for this purpose: at each BO iteration, we generate a large set of candidate architectures, and discard those without at least one favorable motif. As such, the BO agent now only needs to select from a smaller subset of architectures deemed more promising, thereby “warm starting” the search on a new task.
>
> [Discarding bad motifs is risky] Ultimately, the trade-off here is whether the efficiency gained from constraining the search space justifies the risk of missing out better solutions in the discarded/unexplored region. We argue that it is the case in NAS where the search space is immense (DARTS [3] search space with only 4 operation blocks features 4.4e12 combinations) and each query is computationally expensive. Furthermore, experimentally, we show in Fig 1 and 7 that across various search spaces, at least in the experiments we conducted the “good” yet smaller subsets still contain the best candidates among those randomly sampled, and in Fig 4, the final solutions with pruning are not worse than those without. This supports the informativeness of our learnt motifs and we hope this also alleviates the concern about the risk of pre-emptively discarding architectures based on motifs.
>
> (---Continued below---)

---

> > ### Author Response · Authors · 2020-11-19
> > **Response to Reviewer 4 (2/3)**
> >
> > (---Continued from above---)
> >
> > [Bad motifs are not discarded forever]:  The criterion used for pruning which is the average gradient of the motifs, is not static and will be updated as we update the new GP surrogate with more query data. Thus, even if a motif is denoted as unfavourable based on the old-task surrogate and thus discarded at the initial BO iterations, it might become favourable later if its average gradient improves due to positive contributions from the new-task data and surrogate. This dynamic pruning criterion further mitigates the risk of discarding motifs discussed above.
> >
> > 3. **The authors claim that the identified motif is transferable. However, evidence of this claim is only from similar datasets. When is transferring effective and can it harm in some case?**
> >
> > [Dataset choices] Our setting of transferring architecture knowledge learnt from CIFAR-10 to CIFAR100 and ImageNet datasets is the standard approach to assess the transferability in NAS [3-8]. Actually one of the key motivations and selling points of NAS-Bench-201 dataset [7] is that it provides architecture performances on multiple image datasets and thus allows NAS researchers to study the transferability of architectures found.
> >
> > [Why motifs are transferable] the popular cell-based search space in NAS assumes that the good cells are “more likely to generalize to other problems” (quote from [8]) and this assumption has also been empirically verified in many prior works [3-8]. This enables us to perform search on a proxy dataset (e.g. CIFAR-10) and transfer the found cells to a larger dataset (e.g. ImageNet). The motifs in our paper are simply the building blocks of a cell and thus we believe they should also be transferable. In our work, we also empirically demonstrate that the motif information is transferable across different image tasks on NAS-Bench-201 dataset.
> >
> > [When is transferring ineffective] Transfer learning, by definition, often refers to transferring useful knowledge from old tasks to a new **related** task. If the old tasks are poorly correlated or even negatively correlated with the new target task, we believe simple transferring can be ineffective or even harm the search on the new task.
> >
> > [3] Liu, H., Simonyan, K. and Yang, Y., 2018, September. DARTS: Differentiable Architecture Search. In International Conference on Learning Representations.
> >
> > [4] Luo, R., Tian, F., Qin, T., Chen, E. and Liu, T.Y., 2018. Neural architecture optimization. In Advances in neural information processing systems (pp. 7816-7827).
> >
> > [5] Wang, L., Xie, S., Li, T., Fonseca, R. and Tian, Y., 2019. Sample-efficient neural architecture search by learning action space. arXiv preprint arXiv:1906.06832.
> >
> > [6] Shi, H., Pi, R., Xu, H., Li, Z., Kwok, J. and Zhang, T., 2020. Bridging the Gap between Sample-based and One-shot Neural Architecture Search with BONAS. Advances in Neural Information Processing Systems, 33.
> >
> > [7] Dong, X. and Yang, Y., 2020. Nas-bench-201: Extending the scope of reproducible neural architecture search. International Conference on Learning Representations.
> >
> > [8] Zoph, B., Vasudevan, V., Shlens, J. and Le, Q.V., 2018. Learning transferable architectures for scalable image recognition. In Proceedings of the IEEE conference on computer vision and pattern recognition (pp. 8697-8710).
> >
> > 4. **the gradient-based importance evaluation doesn’t consider the variance of gradients... lacks a discussion on uncertainty**
> >
> > Following the reviewer’s suggestion, we repeat the motif discovery experiments (Fig. 1) by taking into consideration the empirical variance of each feature gradient across architectures. Specifically, we incorporate the uncertainty by  dividing (normalising) the averaged gradient by the empirical standard deviation of the gradient; this essentially penalises those gradient estimates with high variability and thus uncertainty. Note that an alternative way to incorporate variance is to add it to the averaged gradient. However, such additive form often involves a trade-off parameter (e.g. like beta in GP-UCB) to balance the effect of the two terms. Such a trade-off parameter requires careful tuning and thus we didn’t investigate this alternative for our rebuttal. While the main conclusions of our approach remain unchanged and results on N201 search space remain almost the same, we find that the gradient-variance-aware motif selection improves the results on the DARTS search space (right plot in Fig. 1 b)) where we have small number of training data; the gap between the median of good-arch distribution (green dash) and that of all-arch distribution (blue dash) gets wider compared to our original plot. Thus, the consideration of uncertainty may be useful during the early phase of the search when we have a less accurate surrogate. The figures and the description in Sec. 3.2 have been updated to reflect this change.
> >
> > (---Continued below---)

---

> > > ### Author Response · Authors · 2020-11-19
> > > **Response to Reviewer 4 (3/3)**
> > >
> > > (--- Continued from above---)
> > >
> > > One point that we would like to clarify: referring to the reviewer’s remark that ‘this paper lacks the kind of discussion on uncertainty’, we **do** use uncertainty information to drive evaluation selection through BO. In fact, we believe the well-calibrated uncertainty estimate by the GP is a key factor differentiating NAS-BOWL over previous approaches with non-GP surrogates such as BANANAS.
> > >
> > > 5. **the strategies are described for each datasets separately. I couldn't find general procedures for the architecture search from the main text of the paper.**
> > >
> > > We agree that showing the general procedures in the main text is helpful, and we have moved our method’s algorithm  from App. A to Algorithm 1 in the main text.
> > >
> > > [Generality of approach] We would like to emphasise that, for example, the setup for our search strategy NAS-BOWL is **identical** on NAS-Bench-101 (a closed-domain task) and DARTS (a popular open-domain task) search spaces, even though they are very different in the way networks are represented, sizes of search space and number of allowed types of operations. The only differences in handling different search spaces are in our pre-processing wrapping steps (e.g. converting edge-attributed to node-attributed graphs) used to standardise the network representation; this is to enable direct application of our search strategy and fair comparison with other baseline methods. Also the consistently superior performance of our NAS-BOWL on different search spaces is further evidence of the generality of our approach. Lastly, we would like to emphasise that NAS-BOWL is actually near hyperparameter-free, as the only new hyperparameter introduced (number of WL iterations H) is automatically tuned by optimising the GP log-marginal likelihood. We also show that NAS-BOWL is robust to other existing hyperparameters common to all BO methods (e.g. choice of acquisition function, pool size of number of architectures to generate) in App. F.3.
> > >
> > > 6. **Overall, the idea would be reasonable, and the approach would be useful. But the standard WL kernel is directly used and a gradient-based importance evaluation is also known**
> > >
> > > Although we acknowledge the point made by the review, we would like to point out that the main novelty of our work does not lie in proposing a new graph kernel, but in leveraging the well-known WL kernel for the specific task of NAS via a Bayesian optimisation approach (R3); the choice of the kernel enables us to naturally handle the graph structure of neural architectures where achieving query-efficient optimisation performance (R1, R2). We would also emphasise that another main novelty of the work is the interpretability provided by our approach, which to the best of our knowledge represents one of the first attempts in the NAS literature. As acknowledged by R3, a general graph kernel only measures similarity between graphs and does not necessarily perform subgraph selection, so the utilisation of the WL kernel is novel and nontrivial in our task. Specifically, our use of the WL propagation procedure (Fig. 5) in combination with the surrogate derivatives in the NAS context leads to the extraction of  interpretable graph features responsible for good architecture performance.

---

> > > > ### Comment · AnonReviewer4 · 2020-11-19
> > > > **About expectated gradient**
> > > >
> > > > Thank you for exhaustive responses. I still couldn't fully follow them but mostly they seem reasonable. One thing I'd like to quickly clarify is as follows.
> > > >
> > > > > One point that we would like to clarify: referring to the reviewer’s remark that ‘this paper lacks the kind of discussion on uncertainty’, we do use uncertainty information to drive evaluation selection through BO.
> > > >
> > > > My claim is that the original expected gradient based pruning does not consider uncertainty of GP.
> > > >
> > > > According to the manuscript, gradient information is also used for pruning in the exploration (not only for 'explaining').
> > > >
> > > > However, pruning motifs by the expected value of gradient does not incorporate uncertainty information, obviously.
> > > >
> > > > In my understanding, if this expected gradient pruning is applied to the initial GP without any observations (with constant prior mean and unit variance), all the motifs may be pruned because their expected gradient would be 0.
> > > >
> > > > In this sense, for the pruning purpose, dividing by standard deviation is not convincing for me (for interpretation purpose, it would be ok). This strategy seems to further tend to be biased to exploitation.
> > > >
> > > > Sorry if I overlook something.
> > > >
> > > > Best

---

> > > > > ### Author Response · Authors · 2020-11-19
> > > > > **Thank you for your comment.**
> > > > >
> > > > > Many thanks for your prompt reply! Here is our response:
> > > > >
> > > > > 1. **According to the manuscript, gradient information is used for pruning in the exploration ...However, pruning motifs by the expected value of gradient does not incorporate uncertainty information**
> > > > >
> > > > > Thank you for raising this issue. We would like to clarify that our use of motif gradient information for pruning is not intended to encourage exploration but **purely for better exploitation** (On this, if there is any sentence/phrasing in our paper that causes this confusion, we would be grateful if you could point it out so that we can fix it). One good analogy to our pruning practice is the construction of the trust region in TurBO* [1], a SoTA high-dimensional BO method for continuous spaces: In TurBO, a trust region is constructed around the best data point observed so far to constrain the large search space to a small sub-region deemed promising, and thus mitigate the problem of BO over-exploring in high-dimensional space. We argue that we have the same problem in NAS:  firstly,  under the transfer learning setup where we apply pruning, we have learnt useful motifs from the previous tasks. Therefore, we do not have to over-explore because we can exploit the useful motif knowledge to constrain the search space to a promising  “trust region” to warm start on the new task. Secondly, as discussed, the search space of NAS is typically huge and each evaluation is expensive (especially so on the transferred task), and the key objective is to converge to promising solutions quickly so over-exploration is clearly undesirable. Meanwhile, we acknowledge that pruning the candidate architecture pool used for optimising the acquisition function is only one of the potential ways to exploit such motif prior knowledge.
> > > > >
> > > > > In addition, similar to the trust region in TurBO, pruning is **not** a replacement of the exploration-exploitation trade-off that occurs at the next step (Line 13 in Algorithm 1) where we evaluate the acquisition functions to select which candidate locations to be evaluated; there we use Expected Improvement, which **does** consider the GP’s uncertainty. The key procedure is summarised below and more formally in Algorithm 1:
> > > > >
> > > > > **Without pruning**:
> > > > > Generate a pool of candidate architectures ----> Select top-k candidates by EI (exploitation & exploration)-----> Evaluate selected cells (costly)
> > > > >
> > > > > **With pruning using transferred motif knowledge**:
> > > > > Generate a pool of candidate cells ----> Prune the pool (exploitation) ----> Select top-k candidates by EI  (exploitation & exploration)---> Evaluate selected cells (costly)
> > > > >
> > > > > *: We use TurBO for illustration -- there are other BO methods that also feature some sort of search space constraining to improve performance. Other examples include [2] & [3].
> > > > >
> > > > > [1] Eriksson, D., Pearce, M., Gardner, J., Turner, R.D. and Poloczek, M., 2019. Scalable global optimization via local bayesian optimization. In Advances in Neural Information Processing Systems (pp. 5496-5507).
> > > > >
> > > > > [2] Wang, L., Fonseca, R. and Tian, Y., 2020. Learning Search Space Partition for Black-box Optimization using Monte Carlo Tree Search. Advances in Neural Information Processing Systems, 33.
> > > > >
> > > > > [3] Wang, Z., Gehring, C., Kohli, P. and Jegelka, S., 2018, March. Batched large-scale bayesian optimization in high-dimensional spaces. In International Conference on Artificial Intelligence and Statistics (pp. 745-754). PMLR.
> > > > >
> > > > > 2. **if this expected gradient pruning is applied to the initial GP without any observations (with constant prior mean and unit variance), all the motifs may be pruned**
> > > > >
> > > > > We believe such an extreme scenario would never occur in our method. Note that in Eq 3.4, the averaged gradients are averaged over observed data points. There would be no averaged gradient when we have no observations.
> > > > >
> > > > > Moreover, **we only introduce pruning in the simple transfer learning setting** where the motif gradients are computed using the GP surrogate trained on the query data of a previous task, as well as that trained on the non-empty query data of the current task. We will never obtain a pruning criteria with an untrained prior GP.
> > > > >
> > > > > As always, if there is anything else that remains unclear, we'd be more than happy to clear any doubts and concerns you might have.

---

> > > > > > ### Comment · AnonReviewer4 · 2020-11-19
> > > > > > **Thank you for clarification**
> > > > > >
> > > > > > Thank you for your quick response.
> > > > > >
> > > > > > > **purely for better exploitation** (On this, if there is any sentence/phrasing in our paper that causes this confusion, we would be grateful if you could point it out so that we can fix it)
> > > > > >
> > > > > > At least in the first review, from the description in the manuscript, I couldn't find that the pruning is for pure exploitation and it is an analogy to trusted region approach. Even if it is a trusted region based warm start, elaborated descriptions would have been required to justify the transfer based motif pruning. As the authors mentioned in the response, transfer can be harm, i.e., not only for over-exploration, over-exploitation can also happen, and narrowing the search space is a still controversial topic in BO. As I commented in the first review, descriptions on the gradient-based pruning procedure for the exploration was quite vague in the first manuscript. However, I think the new Algorithm 1 is really informative. It clarifies what 'pruning rule' is actually designed by the authors, which I didn't find in the first manuscript as a general procedure though it should contain essential information about proposed framework.
> > > > > >
> > > > > > > We believe such an extreme scenario would never occur in our method.
> > > > > >
> > > > > > Of-course, I know this is an extreme case which does not happen in practice, but I believe this toy example clarifies the underlying implication of the average gradient (i.e., effect by ignoring variance).
> > > > > >
> > > > > > > Note that in Eq 3.4, the averaged gradients are averaged over observed data points. There would be no averaged gradient when we have no observations.
> > > > > >
> > > > > > This is a bit confusing for me. The gradient itself should be able to be defined regardless of existence of observed points. I guess a reason of this non-existence is that the 'prior' p(phi^j(G)) is defined by the observed points this time, but it is not essential for the discussion here. My point is only in the average gradient does not incorporate uncertainty, and in the last reply, the author actually admitted that the pruning is purely for exploitation.

---

> > > > > > > ### Author Response · Authors · 2020-11-22
> > > > > > > **Many thanks and this discussion has been helpful**
> > > > > > >
> > > > > > > Many thanks for your reply and we are glad that this discussion, alongside with our updated script, have cleared most of the concerns. For the latest points raised by you, here is our reply:
> > > > > > >
> > > > > > > **i.e., not only for over-exploration, over-exploitation can also happen, and narrowing the search space is a still controversial topic in BO**
> > > > > > >
> > > > > > > We actually very much agree with you on this point -- and that is why we propose to use pruning conservatively only under a transfer learning setup, where we have high-quality prior information to justify the bias toward exploitation. Also note that in prior NAS works, an architecture cell searched on CIFAR10 is often transferred directly (but with more repetitions) to ImageNet without performing any additional search; in this sense our use of motif-based pruning to warm start BO search on the new task is already more explorative. Under the general setup, we purely rely on the BO to do the exploitation-exploration tradeoff and introduce no further narrowing of the search space despite that also being possible. However, we believe a better approach to balance the exploration and exploitation under our NAS-BOWL framework would be an interesting future direction.
> > > > > > >
> > > > > > > **the gradient itself should be able to be defined regardless of existence of observed points**
> > > > > > >
> > > > > > > We’d like to make a quick clarification here: our GP gradients w.r.t the **graph features $\boldsymbol{\phi(G)}$** (i.e. the feature vectors shown in Fig 5) are different from the surrogate gradients w.r.t the inputs (in our case, it will be the derivative w.r.t the input **graphs $G$** themselves, which is ill-defined in the graph input space). The latter, if indeed defined properly, exists regardless of the presence of training points but the former doesn’t because we know the presence of different graph features/motifs only after observing training graph data (you may refer to Fig 5 -- note that the dimension and exact meaning of the feature vector are dependent on what graphs you’ve observed). Therefore, if there is no training point, there is no feature/motif learnt and the derivatives w.r.t motif don’t exist.
> > > > > > >
> > > > > > > Once again, we want to thank you for engaging in this fruitful discussion with us. We sincerely hope you could consider adjusting your assessment if you think that we have addressed your concerns sufficiently; otherwise, we are more than happy to clarify further if you have more questions.

---

> > > > > > > > ### Comment · AnonReviewer4 · 2020-11-22
> > > > > > > > **Thank you for patient reply**
> > > > > > > >
> > > > > > > > Thank you for your response. I really appreciate your series of patient replies. I also feel the discussion is really fruitful.
> > > > > > > >
> > > > > > > > > and that is why we propose to use pruning conservatively only under a transfer learning setup, where we have high-quality prior information to justify the bias toward exploitation. Also note that in prior NAS works, an architecture cell searched on CIFAR10 is often transferred directly (but with more repetitions) to ImageNet without performing any additional search; in this sense our use of motif-based pruning to warm start BO search on the new task is already more explorative. Under the general setup, we purely rely on the BO to do the exploitation-exploration tradeoff and introduce no further narrowing of the search space despite that also being possible.
> > > > > > > >
> > > > > > > > The above statement clarifies the stance of the proposed method, but it is seemingly a heuristic/empirical justification from a limited number of well-known datasets. For example, in general, it seems difficult to know that you have 'high-quality prior information to justify the bias toward exploitation' specific to a given new task which has been unseen before (unlike known tasks, quality of motifs is difficult to know beforehand), except for the case you have enough knowledge about a new task (if I overlook some discussion over this issue in the paper, I'd appreciate if you could tell it). Therefore, for me, how the balance of the exploit-exploration in the entire proposed procedure is qualitatively or theoretically justified is still a bit vague. Of-course, describing heuristically good settings for typical datasets would be valuable (in particular for ICLR), but in a perspective of the proposal of the new exploration strategy for NAS in general tasks, technical novelty is seemingly somewhat limited, which is actually one of my intentions in the first comment.
> > > > > > > >
> > > > > > > > > Therefore, if there is no training point, there is no feature/motif learnt and the derivatives w.r.t motif don’t exist.
> > > > > > > >
> > > > > > > > As the author stated the dimension corresponding to an unseen motif is not created by the procedure in Fig.5. However, it does not mean that the motif does not exit in the search space. In other words, WL only implicitly removes the dimension phi^j(G) = 0 for the given data. Thus, I still think that the expected derivative can be defined (probably, as 0), though the existence itself is not essential for my claim that is the average gradient does not consider uncertainty.
> > > > > > > >
> > > > > > > > Best

---

> > > > > > > > > ### Author Response · Authors · 2020-11-23
> > > > > > > > > **Response and clarification of the main contributions**
> > > > > > > > >
> > > > > > > > > Many thanks for your patient and active discussions.
> > > > > > > > >
> > > > > > > > > At this point, we think it is helpful to re-emphasise the key contributions of our work over which this discussion evolves: we introduce the concept of interpretability into NAS by showing that interpretable network motifs/features can help explain network performances, and that they can be extracted by our NAS strategy and leveraged for practical uses. We believe this very contribution,  as far as we know, is the **first** in the NAS community and we are grateful that all other reviewers (R1, R2, R3) agree on this novel aspect of our work.
> > > > > > > > >
> > > > > > > > >  Throughout our discussion, it seems that this contribution has somewhat been taken for granted (for example, we feel most of the remarks on motif-based pruning do ground on the acknowledgement that motifs are somewhat interesting) and in fact all experiments, around which most of our discussion revolves, are just there to **empirically verify our main claim that interpretable motifs are useful and interesting**. This includes the qualitative results (e.g. how different NAS strategies agree implicitly on finding similar substructures and how motif discovery could potentially lead to new insights in NAS) and quantitative results (e.g. Fig 1 and the **transfer learning/pruning** experiments discussed here).
> > > > > > > > >
> > > > > > > > > Moreover, the approach we proposed to extract these motifs also requires careful thinking and novel integration between graph kernels and GP literatures. As pointed out by R3, a general graph kernel only measures similarity between graphs and does not necessarily perform subgraph selection. On the other hand, simply using GP surrogate derivatives, if not combined with our novel use of the WL propagation procedure in the NAS setting, also wouldn’t lead to the discovery of good and bad network motifs. Although we’ve empirically shown that this approach as well as our method for applying interpretable motifs in transfer learning indeed work well, we acknowledge again that they are not the only way towards interpretable NAS. If the reviewer has further suggestions on this, we will be glad (and grateful) to incorporate them into the paper as future directions and more importantly, as we mentioned in the general reply to all reviewers, we hope our work would spur further research interests on interpretability in NAS.
> > > > > > > > >
> > > > > > > > > ----------------------------------------------------------------------------------------------------------
> > > > > > > > > With this in mind,  we would also like to respond to some of the reviewer’s specific points:
> > > > > > > > >
> > > > > > > > > >it seems difficult to know that you have 'high-quality prior information to justify the bias toward exploitation' specific to a given new task which has been unseen before (unlike known tasks, quality of motifs is difficult to know beforehand), except for the case you have enough knowledge about a new task (if I overlook some discussion over this issue in the paper, I'd appreciate if you could tell it)
> > > > > > > > >
> > > > > > > > > When we transfer the motifs learnt from CIFAR10 to CIFAR100 and ImageNet, we treat these two new tasks as ”unknown beforehand” and never assume the motifs learnt from CIFAR10 definitely work on the new tasks (we never have "enough knowledge about a new task“). In fact, it is the empirical evidence on these new tasks that verifies our hypothesis that good motifs are transferable across different tasks. We have also already provided a short, qualitative explanation on why motifs transfer well in both the first response and in Sec. 3.2, Page 6 (about relating back to the original motivation of cell-based search).
> > > > > > > > >
> > > > > > > > > Furthermore, at the very least, even in the unusual case that new and old tasks are poorly correlated therefore motifs learnt on old don’t transfer well to new tasks, our method will still suffer less than the common practice of transferring the entire cell learnt and **not performing additional searching on the new task at all**, because 1) as we said previously, our method still perform BO search on the new task and 2) we dynamically update the motif-based pruning rules with observations of the new task to mitigate the problem.

---

> > > > > > > > > > ### Comment · AnonReviewer4 · 2020-11-23
> > > > > > > > > > **Thank you again for the reply**
> > > > > > > > > >
> > > > > > > > > > Thank you for your reply. Really sorry for being persistent, but I still have a few points to clarify.
> > > > > > > > > >
> > > > > > > > > > > qualitative results
> > > > > > > > > >
> > > > > > > > > > My intention behind the word 'qualitative' in the previous comment is a non-empirical justification of the exploration strategy (with gradient-based pruning) such as a guarantee that the algorithm can explore the entire search space with finite time or the interpretation as the optimization through some well-established criterion, and so on. In this sense, the proposed search algorithm is currently a sensible heuristics that is not fully justified qualitatively (at this point, I describe below again).
> > > > > > > > > >
> > > > > > > > > > > When we transfer the motifs learnt from CIFAR10 to CIFAR100 and ImageNet, we treat these two new tasks as ”unknown beforehand” and never assume the motifs learnt from CIFAR10 definitely work on the new tasks (we never have "enough knowledge about a new task“). In fact, it is the empirical evidence on these new tasks that verifies our hypothesis that good motifs are transferable across different tasks. We have also already provided a short, qualitative explanation on why motifs transfer well in both the first response and in Sec. 3.2, Page 6 (about relating back to the original motivation of cell-based search).
> > > > > > > > > >
> > > > > > > > > > My point is that people already know how similar CIFAR10, CIFAR100, and ImageNet, and know how transferable they are. When you have a completely new task, I still think it is not easy to determine the past motif is trasferable in principle (even among related tasks). The authors referred several prior works about transferability in the first reply, but I don't think just quoting 'more likely to generalize to other problem' from the past paper is enough to justify the full exploitation motif transfer as a general warm start methodology of BO-based NAS. At least, to clarify the applicable scope of the proposed method, it should have been described in more detail in the main text.
> > > > > > > > > >
> > > > > > > > > > > Furthermore, at the very least, even in the unusual case that new and old tasks are poorly correlated therefore motifs learnt on old don’t transfer well to new tasks, our method will still suffer less than the common practice of transferring the entire cell learnt and **not performing additional searching on the new task at all**, because 1) as we said previously, our method still perform BO search on the new task and 2) since we are dynamically updating the motif-based pruning rule with new task data and surrogate to mitigate the problem.
> > > > > > > > > >
> > > > > > > > > > First, I don't think it is unusual. The negative transfer (the situation that transfer harms) is a common phenomenon. In general, for a new task, it is not trivial to know how strongly correlated to the past data, beforehand. Second, the claim in 1) and 2) are seemingly reasonable intuitions, but as I partially mentioned in the beginning of this reply, it would not be more than intuition. Claiming advantage based on the empirical evaluation is of-course ok, but again, in a perspective of technical significance, for me, the proposed search algorithm is not particularly strong and appropriateness is not fully clear (on the other hand, I admitted the practical usefulness from the first review, and therefore, my evaluation is 'borderline'), and another concern which I couldn't have discussed so far is in the interpretation of the gradient-based importance evaluation.
> > > > > > > > > >
> > > > > > > > > > > we marginalise over all architectures containing the motif and obtain a global measure of importance regarding the motif feature. In fact, the integration or averaging of some sort over local gradients, as we did, is also an approach taken in many principled gradient-based global attribution methods, such as Occlusion-1 [1] and Integrated Gradient [2], to analyse the contribution of input features to the model prediction.
> > > > > > > > > >
> > > > > > > > > > For me, the interpretation of integrating the gradient in the entire feature space for importance evaluation is not clear. For example, if phi^j(G) has both of positive and negative gradients in the different values of phi in the feature space, then the integral can be 0 in total (note that this is different discussion from one we've discussed so far which was about expectation over predictive distribution of GP, the current discussion is in the integral over the feature space), but it is not clear whether this motif is important or not. The authors showed two related papers in [Local vs Averaged Gradients]. However, for me, above issue is unclear.
> > > > > > > > > >
> > > > > > > > > > Best

---

> > > > > > > > > > > ### Comment · AnonReviewer4 · 2020-11-24
> > > > > > > > > > > **summary of my understanding**
> > > > > > > > > > >
> > > > > > > > > > > Since my previous reply was a bit messy, I summarize current my understanding as follows. Sorry for repeated postings.
> > > > > > > > > > >
> > > > > > > > > > > [ lack of guarantee for pruning ]
> > > > > > > > > > >
> > > > > > > > > > > The proposed pruning strategy in Algorithm 1 restricts search space based on gradient of current (posterior mean of) GP at each iteration. This is seemingly reasonable, but there's no theoretical guarantee that the exploration algorithm does not overlook important motifs in the entire procedure (the gradient is updated every iteration and new candidate is selected by EI, but it does not immediately resolve the possible risk (at least, not trivial)). Restricting search space in the acquisition function maximization is an important issue on BO, and therefore, more rigorous justification would make the paper more attractive. Further, for this pruning issue, the two concerns below are also closely related.
> > > > > > > > > > >
> > > > > > > > > > > [ motif transfer is source-target relation agnostic ]
> > > > > > > > > > >
> > > > > > > > > > > The motifs with large gradient are transferred without evaluating similarity between source and target tasks. In transfer learning, how strongly transferring the past information to a new task is an essential issue. Transferring everything (top gradient) always (any target data) is seemingly too adventurous and is not technically attractive as a novel warm-starting strategy. At least, careful discussion would have been preferred to clarify applicability of the proposed framework because the negative transfer is also possible to happen.
> > > > > > > > > > >
> > > > > > > > > > > [ interpretability of integrated gradient ]
> > > > > > > > > > >
> > > > > > > > > > > The interpretability is a central claim of the paper, but I don't think the interpretability of the gradient-based importance evaluation is discussed enough. The gradient is integrated in the entire feature space, and then, it is difficult to find importance. For example, when the gradient has both positive and negative values in the space, the resulting integrated value may be 0. In this case, importance of this motif is not clear, and further, meaning of comparing different motifs based on this integrated value becomes also unclear because of this property.
> > > > > > > > > > >
> > > > > > > > > > > Best,

---

> > > > > > > > > > > > ### Author Response · Authors · 2020-11-25
> > > > > > > > > > > > **Final Response and Summary of Our Understanding (1/2)**
> > > > > > > > > > > >
> > > > > > > > > > > >
> > > > > > > > > > > > We would like to thank the reviewer once again for engaging with us during the rebuttal. Here is our final response -- we also made a final update to our manuscript to add discussions as requested. We hope this further clears things up.
> > > > > > > > > > > >
> > > > > > > > > > > > First of all, for the first two comments ([lack of guarantee for pruning] and [motif transfer is  source-target relation agnostic]), we would like to reiterate the point made from the previous response again as it still seems that our core contribution has been somewhat assumed (that interpretable motifs are important and interesting). We would like to emphasise that **this paper is not primarily about proposing the perfect transfer learning/pruning method** which, as we have repeatedly clarified, is a working example to show motifs can be used in a practical way. We have updated our paper to make this even clearer. In fact, while we do think the pruning experiment provides important empirical evidence to support our main thesis (*we  think the reviewer also acknowledges the empirical result, at least on datasets standard to almost all NAS works*), the main points of our paper stand, by and large, **with or without** this particular part. **We hope our main contributions are properly taken into account, as we believe even the criticisms of the transfer learning setting that are somewhat central to the reviewer’s assessment are grounded on the implicit acknowledgement that motif discovery, interpretation and the application are useful. To the best of our knowledge, these are all novel to this work and should not be assumed to be known**.
> > > > > > > > > > > >
> > > > > > > > > > > > > [Lack of Guarantee for Pruning]
> > > > > > > > > > > >
> > > > > > > > > > > > We thank the reviewer and agree this will be an interesting and critical future work both for us and the entire NAS community, which is somewhat weak on theory as a whole, and we added a discussion in Conclusion. Nonetheless, we’d just like to remind the reviewer that the present paper is a **practical** one: we hope the reviewer agrees with us that good empirical performance especially under limited budgets and extreme costs takes a higher priority over whether the algorithm converges *eventually*, at least in *this* paper.
> > > > > > > > > > > >
> > > > > > > > > > > > > [ motif transfer is source-target relation agnostic ]
> > > > > > > > > > > >
> > > > > > > > > > > > We acknowledge the point made -- we add a discussion in Sec 5, Pg 9 below Table 1 to qualify our claim as requested. We’d still like to remind the reviewer that 1) the claimed ‘lack of technical novelty’ of pruning remains conditional on our major novelty of discovering and utilising the interpretable motifs, 2) unconstrained search on expensive target task is infeasible by setting (otherwise we search directly instead of “transfer”), and “constraining search space” or some sort of bias towards exploitation here is not controversial, but essential and 3) as the reviewer has somewhat acknowledged, the most popular, “full-exploitation” alternative of transferring cells directly across tasks suffers even more from **all** their criticisms here and in the first point.
> > > > > > > > > > > >
> > > > > > > > > > > > > [Interpretability of averaged gradient]
> > > > > > > > > > > >
> > > > > > > > > > > > We believe this remark is more related to a core contribution of this paper, and we thank the reviewer for this. First, we find it a bit strange when the reviewer says “it is difficult to find importance/meaning of comparing motifs is unclear”, when we clearly show our method **does** find importance in NAS-Bench-101, all 3 tasks of NAS-Bench-201 and DARTS search space (Fig 1 and 7) -- if we can't find importance/distinguish different motifs, none of the results we've shown would've been possible. Second, the example the reviewer gives is **not** a case when averaged gradient **fails**. On the contrary, it is exactly an example of when averaged gradient **works**.
> > > > > > > > > > > >
> > > > > > > > > > > > ( -- to be continued below -- )

---

> > > > > > > > > > > > > ### Author Response · Authors · 2020-11-25
> > > > > > > > > > > > > **inal Response and Summary of Our Understanding (2/2)**
> > > > > > > > > > > > >
> > > > > > > > > > > > >
> > > > > > > > > > > > > A motif with high and diverse local gradient magnitudes but average to near-0 is not important for the purpose of interpretability, as it **doesn’t consistently explain the network performance by itself** (just based on such motifs, one cannot conclusively deduce the impact on performance of an arbitrary, unseen architecture **in general**), and this is not a bug, but a feature of our design (note that in addition to averaged gradient, we explicitly penalise those gradient estimates with high variance). When the case described by the reviewer occurs it likely is suggesting that the feature is a common part of higher-order motifs that more conclusively correlate the network performance in opposite directions. To give an example, the node of conv1x1 itself as a motif has both positive and negative gradients in different values of phi and even the same value of phi across different training samples, but, as just an illustrating example, it could be because when it appears with conv3x3., there is often better performance but when it appears with maxpool, there could often be worse performance. Presence of conv1x1-conv3x3 and conv1x1-maxpool in this case is far more interesting and valuable in interpreting the network performance, whereas the presence of just conv1x1 has little predictive value and is a confounding factor preventing us from uncovering the truly interesting patterns — this is exactly reflected in the normalised averaged gradient (conv1x1 will have gradient near 0, whereas conv1x1-conv3x3 will have very positive and conv1x1-maxpool will have very negative gradients).
> > > > > > > > > > > > >
> > > > > > > > > > > > > We feel it is also helpful to clarify the difference between **local** vs **global** attribution which seems to cause some confusion. Here we are interested in the latter. The difference is well-elaborated in pg 6-7 of [1] https://openreview.net/pdf?id=Sy21R9JAW, and as the authors of [1] wrote in pg 7, *“global attributions should be used to identify the marginal effect that the presence of a feature has on the output, which is usually desirable from an explanation method.”* -- this is exactly what we do by integrating over local gradients to obtain the “marginal effect”, and we aim to answer the question of “**generally**, what information can human designers tell of the performance based on the presence of certain motifs?” (c.f. Local attribution in our context would be: based on a **specific** architecture, how do we modify it to obtain better performance? There we should use local gradients at that architecture; this can be an interesting direction and even another case to concretely apply interpretable motifs in addition to the transfer learning example, but that’ll be a **distinct** application).

---

### Author Response · Authors · 2020-11-19
**Response to all reviewers**

We would like to thank all reviewers for their valuable suggestions to improve this paper and their appreciation of our proposed method, and we have reflected many valuable suggestions made by the reviewers in the updated version of our paper. We would like to reiterate our key contributions before responding to each reviewer individually.

At a high level, our method uses Bayesian optimization with Weisfeiler-Lehman graph kernels for the task of NAS, thus naturally handling the directed acyclic graph representation of neural architectures. We also show via extensive experiments that our method attains state-of-the-art performance on a wide range of open- and closed-domain tasks with strong data efficiency. We are thankful that in this aspect, our work has been appreciated as “innovative and natural” (R1), “well-motivated” (R2), “convincing” (R3) and “reasonable and useful” (R4)  by all reviewers.

Another major contribution is that we leverage the interpretable motifs extracted during the WL comparison process in combination with their corresponding surrogate gradients to introduce the notion of interpretable NAS. To the best of our knowledge we are the first in doing so, and we are glad that most reviewers reached a consensus on the novelty and usefulness of such interpretability (R1, R2, R3). As a concrete application of interpretability, we also propose and briefly investigate motif-based transfer learning in NAS context in our paper. We hope that our work will inspire further works in this exciting new direction.

---

### Decision · Program_Chairs · 2021-01-07
**Final Decision**

**Decision:**

Accept (Poster)

**Comment:**

Most reviewers found the method proposed to be technically sound, well-motivated and particularly interesting due to the interpretability of its results. Indeed, the extraction of interpretable motifs from NAS is a valuable contribution. One of the reviewers was particularly concerned by the lack of guarantees of the proposed method and a perceived failure mode of averaged gradients. We thank both the reviewer and the authors for the detailed discussion on these points. Ultimately, the benefits of the method proposed and the magnitude of the contributions in the paper outweigh these concerns.